# Use Bayesian Paired Tests with a ROPE to Improve the Comparison of Machine Learning Models

## Abstract

This tutorial paper argues that model comparison in machine learning can be much improved by using *paired testing*, i.e. comparing the predictions of methods A and B on each (common) test example. Due to the limitations of null hypothesis significance testing, a Bayesian approach is recommended, including the use of the region of practical equivalence (ROPE; Kruschke 2015a; Kruschke and Liddell 2018; Benavoli, Corani, Demšar, and Zaffalon 2017). We discuss a Bayesian *t*-test and a Bayesian McNemar test for comparisons on a single task, and Bayesian hierarchical models for comparisons over multiple tasks. Three worked examples are presented to illustrate the methods, and the use of reporting guidelines is discussed as a potential means of changing current practice.

In machine learning we often wish to compare two or more different models, e.g., comparing a new method with some standard models. This is a very common task in machine learning, but despite this very few textbooks in the field give advice on how to do this, and in our experience this comparison is often badly done. One may observe highlighting of the best performing method without any regard for uncertainty. Or error bars may be presented on a per-method basis, ignoring the benefits of paired comparisons. It is ironic that the ML field develops very sophisticated statistical models for its tasks, but often fails to use good statistical practice when doing model comparison.

The aim of this paper is to help improve this situation. The first improvement is to use *paired testing*, comparing the results of methods A and B on each individual test example. Standard frequentist methods include paired *t*-tests and McNemar's test, as discussed below in sec. 5.1. However, these are inadequate due to the limitations of point null hypothesis significance testing in frequentist statistics (see, e.g., the American Statistical Association 2016 statement on *p*-values, Wasserstein and Lazar 2016), as discussed in sec. 5.2. An example of a point null hypothesis is that the expected difference $\Delta$ in the loss between methods A and B is exactly 0. Although there are Bayesian methods that also make use of point null hypotheses, we argue that the hypothesis that $\Delta$ is exactly zero is rarely reasonable.

Dispensing with the point null hypothesis, we make use of a *region of practical equivalence* (ROPE, Kruschke 2015a; Kruschke and Liddell 2018; Benavoli, Corani, Demšar, and Zaffalon 2017), which defines a region where the methods are judged to be practically equivalent. In general the use of the ROPE is termed *equivalence testing*, and this can be developed under both frequentist and Bayesian analyses, see, e.g., Wellek (2010). In sec. 5.3 we discuss equivalence testing further, and argue for the merits of Bayesian methods.

The above discussion covers the comparison of two methods on a single task. The standard method for comparing two or more methods on multiple tasks is the Friedman test, as discussed by Demšar (2006). The Bayesian alternative to this is to build a hierarchical model (again including the ROPE), with the performance on each task being modelled as drawn from from a shared prior distribution.

We are not the first to advocate for the Bayesian+ROPE approach. Notably Benavoli et al. (2017) did so, but their work mainly focused on the comparison of *learning algorithms*, using cross-validation to take into account the effect of training set variability. While this is very worthwhile, we believe that it is more germane to focus on the comparison of *predictors* trained on a given development set, and evaluated on a common test set, as this is the setup generally used for benchmarks. (See sec. 1 for elaboration of this point.) Note also that Benavoli et al. (2017) (henceforth BCDZ) used a cross-validation approach, but used comparisons of the

*average* performance on each cross-validation test fold, not on each individual test example. In comparison to BCDZ we also highlight issues of fixed and random factors and experimental design (sec. 3), and discuss a ROPE analysis of the Bayesian McNemar test (including a hierarchical version). This paper thus fits with the TMLR scope criterion "accounts of applications of existing techniques that shed light on the strengths and weaknesses of the methods".

The structure of this paper is as follows: in sec. 1 we discuss the various kinds of model comparison problem, and sec. 2 reviews loss functions used to assess performance. Sec. 3 considers the general question of experimental design, including fixed and random factors, and sec. 4 reviews the relevant literature. Sec. 5 covers standard frequentist tests for model comparison; criticisms of point null hypothesis significance testing; equivalence testing; and details of a Bayesian *t*-test and a Bayesian McNemar test for comparisons on a single task, and Bayesian hierarchical models for comparisons over multiple tasks. 3 worked examples are presented in sec. 6 to illustrate the methods. Sec. 7 provides a checklist for best practice, and we conclude with a discussion in sec. 8.

# 1 What is the question we wish to answer?

Dietterich (1998) provides a nice taxonomy of questions that we may seek to answer. Firstly, is the focus on a single task, or on multiple tasks? In much applied research, a single task is of interest. But in more core machine learning research, the goal is to find learning algorithms that work well across a range of tasks.

Within a single task, the next question is if we are interested in comparing *predictors* or *learning algorithms*? In supervised learning we have input-output pairs $(x, y)$. A predictor takes in an input $x$, and makes a prediction for its associated output $y$; this may be a classification label, a real-valued prediction, or some other kind of output. In contrast, a learning algorithm takes in a dataset of input-output pairs, and constructs a predictor. Benchmark problems like ImageNet (Deng et al., 2009) are usually concerned with comparing predictors which have been trained using a development set. This set is used for model training and selection,[1] and the resulting predictor is then applied to the previously-unused test set to assess its performance.

Multiple tasks can arise from a set of related problems drawn from a distribution over tasks; this is termed a *context* by Lacoste et al. (2012). An example is the 1000 object classes in the original ImageNet database.

If there are multiple predictors or learning algorithms, it is also necessary to consider exactly what comparisons are being made. If we are running a contest, we may care about establishing if the best performing method is significantly better than the others. If we are proposing a new method, we may wish to establish if its performance is better, similar or worse than each of the other competitor methods.

We summarize the different questions as:

Q1: Comparison of predictors on a single task;

Q2: Comparison of learning algorithms on a single task;

Q3: Comparison of learning algorithms across multiple tasks.

Our main focus below is on Q1 and Q3, as these are most related to the comparison of performance on benchmark problems.

# 2 Assessing predictions: Loss

When making comparisons, it is important to consider what is an appropriate measure of the loss of a predictor, based on the difference between the prediction made for an input value $x$ and the observed value $y$. For example in binary classification problems, one can obtain a "hard" prediction for each test example (e.g., by thresholding a probabilistic output at some value), and use the 0/1 loss, i.e. having zero loss if the labels agree, and 1 otherwise. But it may be important to consider the relative importance of false positives

---

[1]The development set may be split into training and validation parts in order to carry out model selection.

and false negatives, ascribing different losses to these situations, and setting the threshold to minimize the expected loss (a.k.a. risk). Another alternative is to use the log loss, $-\log p_{\text{model}}(y|x)$. Or one could summarize the probabilistic results on a test set using a receiver-operating-characteristic (ROC) curve, and assess the quality of the results using the area under the ROC curve (AUROC); note that this implicitly uses all possible thresholds.

For regression problems one commonly considers mean squared error (MSE), root mean squared error (RMSE), mean absolute error (MAE) or log loss. See, e.g., Rainio et al. (2024) for further measures for evaluating predictors on a variety of problems. In all cases it is important to consider which metric best fits with the practical application of the predictor.

Model comparison need not be restricted to supervised learning. For example in an unsupervised learning task one could compare the log losses for models A and B on each test example $x$.

## 3 Factors of variation and experimental design

The problem we seek to address here is to compare two or more predictors or learning algorithms. Note that this kind of question is ubiquitous in science; for example we may wish to compare the efficacy of a new drug treatment for a particular condition against a standard treatment (or placebo), or the yield a new crop variety against an older one. In all cases we seek to *design an experiment* to answer the question. See, for example, Montgomery (2013) for a discussion of experimental design.

In experimental design it is important to distinguish *fixed* and *random* factors of variation. Fixed factors are those of interest to the experimenter, and the experiment assesses specific values of these factors. In contrast, random factors have values that are chosen at random from a large set of possible values, and the aim is to *generalize* to new values of these factors.

If we apply these ideas to Q1, the comparison of predictors, then clearly a fixed factor is the predictor used, and a random factor is the test set consisting of $N$ $(x, y)$ pairs sampled from the distribution $p(x, y)$ that we wish to generalize over. But with methods such as neural networks, there are many other factors that give rise to variation; model architecture, random initialization of the weights, the order of the training examples (if using mini-batch training), weight decay parameters, and so on. For random factors such as weight initialization, we assume (following Rasmussen 1996b, sec. 2.1) that the loss should be defined by *averaging* over such factors. However, for method hyperparameters (such as the architecture or weight decay parameters) we will assume that these have been *optimized*, e.g., by using a validation set within the development dataset.

Another example of fixed and random effects occurs in Q2, the comparison of learning algorithms. Here as well as the fixed effect of the choice of learning algorithm, there is a random effect coming from the different training sets. The choice of test set is also a random factor (as above), but for simplicity we might assume that the common test set has large size, so we can ignore the variability that arises from this source. When comparing learning algorithms, the ideal situation statistically is to have multiple *independent* training sets (of size $n$) available, all drawn from the same distribution $p(x, y)$. This can be hard to achieve if there is limited real-world data without making the training sets very small, but it can be achieved for synthetic data. For real-world data it is tempting to use some form of resampling such as cross-validation or the bootstrap, but this gives rise to complications in the statistical analysis of the results, as the training sets are no longer independent. One example of how to handle this is (frequentist or Bayesian versions of) the correlated $t$-test described in Benavoli et al. (2017). However, note that the correlation parameter $\rho$ needed in this test is not identifiable, and BCDZ make use a heuristic from Nadeau & Bengio (2003) to set it.

For Q3, we will take as fixed effects the different learning algorithms used, but we now consider the set of tasks used as *random effects*, drawn from some distribution over tasks. We then wish to compare the performance of different learning algorithms, generalizing over the choice of specific tasks, to make inferences for the distribution over tasks. We may still have to worry about additional variability arising from training and test sets.

Although Q2 may seem to be simpler than Q3, in fact it may not be easier to answer. The difficulty lies in assessing the variability due to different training sets in Q2 if resampling methods are used, as pointed out above. As Demšar (2006) states "the problem of correct statistical tests for comparing classifiers on a single data set is not related to the comparison on multiple data sets in the sense that we would first have to solve the former problem in order to tackle the latter. Since running the algorithms on multiple data sets naturally gives a sample of independent measurements, such comparisons are even simpler than comparisons on a single data set."

One advantage in machine learning experiments is that we usually have what are termed *repeated measures*,[2] meaning that we can use the same training and test datasets for all algorithms. We can thus compare the losses between two predictors on the same test case; generally such *paired testing* is more powerful, as it removes random variation due to differences in hardness between test cases. See Appendix A for an illustration of this for MSE loss.

A standard linear model that can take into account the fixed and random factors is the 2-way analysis of variance (ANOVA) model:

$$z_{ij} = \mu + \alpha_i + \beta_j + \epsilon_{ij}, \tag{1}$$

see e.g., Montgomery (2013, ch. 13). Here there are two discrete factors, indexed by $i$ and $j$. $z_{ij}$ denotes the loss incurred under the combination $(i, j)$, $\mu$ is the overall mean, $\alpha_i$ denotes the effect of factor $i$ (for $i = 1, \ldots, a$) and $\beta_j$ denotes the effect of factor $j$ (for $j = 1, \ldots, b$), and $\epsilon_{ij}$ is a random error term. It is assumed that $z_{ij} | \mu + \alpha_i + \beta_j \sim N(0, \sigma_\epsilon^2)$. We can consider the case where $i$ indexes a fixed factor, and $j$ a random factor. To ensure identifiability of the parameters, it is imposed that $\sum_i \alpha_i = 0$. The random effects, the $\beta_j$'s, are assumed to be independent and distributed as $N(0, \sigma_\beta^2)$. In general ANOVA is not limited to two factors (it can have one, or more than two) and each one can be fixed or random, but for simplicity we describe the two-factor mixed effects model here. One can also consider not only the "main effects" due to the individual factors, but also interaction terms; for example a term $\gamma_{ij}$ taking on $ab$ values could be included in eq. 1.

One can carry out frequentist or Bayesian inference for the ANOVA model. For example, Montgomery (2013, sec. 13.3) describes describes the estimation of the parameters ($\mu$, the $\alpha_i$'s, $\sigma_\epsilon^2$ and $\sigma_\beta^2$) for the 2-factor mixed effects model. One can also give a Bayesian treatment of this hierarchical linear model, as described, for example, in Gelman et al. (2013, sec. 15.6).

Note that when considering the comparison of only two algorithms, the 2-way ANOVA model can be simplified by considering *paired differences*. For Q1 consider the paired difference $d_j^{AB}$ between predictors A and B on test example $j$. We then have

$$d_j^{AB} \stackrel{def}{=} z_{Aj} - z_{Bj} = \alpha_A - \alpha_B + (\epsilon_{Aj} - \epsilon_{Bj}), \tag{2}$$

where the $\mu$ and $\beta_j$ terms have cancelled out. Note that the last term, which is the difference of two errors that are assumed to be independent and normally distributed, is again normally distributed, but with twice the variance of the individual $\epsilon$'s. One could then use, e.g., a paired $t$-test on these differences to assess if the difference between predictors A and B is statistically significant.

The key issue for determining if a predictor or a learning algorithm is outperforming its competitors is to assess the variability coming from random factors and noise, in comparison to the differences that arise from the fixed effects (such as the identity of the predictor or algorithm).

## 4    Literature review

An early paper on the evaluation of machine learning experiments is the technical report from the DELVE development group in Toronto (Rasmussen et al., 1996), where DELVE is an acronym for Data for Evaluating Learning in Valid Experiments. They considered frequentist and Bayesian analyses of ANOVA models to assess various effects, and also made available datasets and software to carry out testing. Example results are

---

[2]Also referred to as a within-subjects design; for human subjects this means that the same subjects are exposed to different experimental conditions.

available in Rasmussen (1996b). The website `http://www.cs.utoronto.ca/~delve/` still exists, although it appears not to have been updated for many years.

Dietterich (1998) has a nice taxonomy of the questions we wish to answer, as described above in sec. 1. He also includes a question as to whether there is a large or small dataset available. Dietterich's main focus is on choosing between learning algorithms when a small amount of data is available (his question 8), with particular attention to type I errors (detecting a difference when no difference actually exists). For algorithms that can only be run once, he recommends McNemar's test (see sec. 5.1 below), while for algorithms that can be executed ten times he recommends the 5x2cv test (5 iterations of 2-fold cross validation). The latter was subsequently improved by Alpaydın (1999), who developed the combined 5x2cv F test that combines multiple statistics to get a more robust test.

Salzberg (1997) also discusses pitfalls to avoid when comparing algorithms, and finally recommends a *k*-fold cross-validation approach, followed by the binomial (sign) test or McNemar's test to assess statistical significance.

Demšar (2006) focuses on the the statistical comparison of classifiers over multiple data sets. He discusses repeated-measures ANOVA, but recommends instead the Friedman test, due to the strong assumptions necessary for ANOVA. This test is described further in sec. 5.1 under Q3.

Hansen et al. (2011) introduce the concept of the model confidence set (MCS), which is a set of models constructed such that it will contain the best model with a given level of confidence. The algorithm is initialized with all the possible models in the candidate set, and then statistical hypothesis tests are used to compare these models. If the null hypothesis that all models are equivalent in performance is rejected, an elimination rule is used to remove a model from the candidate set, and the hypothesis tests are applied again on the reduced set. This continues until the null hypothesis is not rejected.

The book by Dror et al. (2020) covers parametric and non-parametric frequentist tests. Their Table 4.1 recommends appropriate tests for different evaluation measures. The book is motivated by Natural Language Processing (NLP), although most of the discussion is more general, with only the tasks and evaluation measures being NLP-specific.

The recent paper by Rainio et al. (2024) is concerned with evaluation metrics and statistical tests for machine learning. On the latter topic, the authors present a flow chart (their Figure 3) recommending different statistical tests for different questions and tasks. For example McNemar's test is recommended for comparing binary classifiers, and Friedman's test for comparing models over several test sets. Note that these are for the comparison of predictors rather than learning algorithms.

The tests referred to above are in the frequentist null hypothesis significance testing (NHST) framework. In contrast, one can apply Bayesian methods; this is discussed further in sec. 5. Lacoste et al. (2012) consider Bayesian methods for the comparison of classifiers (using 0/1 loss) on one or multiple tasks. Benavoli et al. (2017) give some Bayesian analyses of experimental results, including a Bayesian correlated t-test, a Bayesian signed-rank test, and a Bayesian hierarchical correlated t-test. These can also be used to compare models on one or multiple tasks. As noted above, their focus is more on Q2, taking into account the effect of training set variability via cross validation.

## 5 Bayesian and frequentist statistical testing

In sec. 5.1 we first discuss standard frequentist point null hypothesis tests for addressing questions Q1, Q2 and Q3 for the comparison of two algorithms. These include the paired *t*-test, McNemar's test, and Friedman's test (for multiple tasks). We also discuss the effect size and its quantification. In sec. 5.2 we outline criticisms of frequentist and Bayesian point null hypothesis testing, which leads us to advocate (in sec. 5.3) the use of a region of practical equivalence (ROPE) and the framework of equivalence testing. We follow this with a discussion of the Bayesian *t*-test (sec. 5.4), a Bayesian McNemar test (sec. 5.5) and Bayesian hierarchical models to handle multiple tasks (sec. 5.6). In sec. 5.7 we discuss the issue of multiple comparisons when comparing multiple predictors. In sec. 5.8 software for carrying out the various tests is described.

### 5.1 Frequentist point null hypothesis tests

**Q1:** For the comparison of predictors, the test to use will depend on the loss function. For a real-valued loss, a standard choice would be a paired $t$-test. We compute the difference of the losses of methods A and B on each test case, to obtain a vector of differences $\mathbf{d} = (d_1, d_2, \ldots, d_N)$. Under the point null hypothesis $H_0$ the mean loss of methods A and B are equal, so that the mean difference will be zero. Under the alternative hypothesis $H_1$ the mean difference is non-zero. The paired $t$-test proceeds by computing a test statistic $t = \frac{m}{s/\sqrt{N}}$, where $m$ is the mean and $s$ the standard deviation of $\mathbf{d}$. A $p$-value for the test statistic can then be computed under the null hypothesis, using a two-tailed test (to reflect that under $H_1$ the mean difference can be greater or less than zero). A small $p$-value will provide evidence to reject the null hypothesis.

In the case of 0/1 losses, we can use McNemar's test (see, e.g., Dietterich 1998). We have predictors A and B, and consider the four different outcomes that can occur on a test case: (00) the example is misclassified by both; (01) the example is misclassified by A but not by B; (10) the reverse of the previous case; and (11) the example is correctly classified by both. Let the counts associated with these four outcomes be $n_{00}$, $n_{01}$, $n_{10}$, and $n_{11}$ respectively, with their sum being $N$, the number of test cases. Under the point null hypothesis the predictors should have the same error rate, so $n_{01} = n_{10}$. The statistic

$$\frac{(|n_{01} - n_{10}| - 1)^2}{n_{01} + n_{10}} \tag{3}$$

approximately follows the $\chi^2$ distribution with one degree of freedom, where the $-1$ term in the numerator is a "continuity correction". Again a $p$-value can be obtained under the null hypothesis.

In some cases the whole test set is used to produce one number, like the area under an ROC curve (AUROC), or the average precision score (AP) for a precision-recall curve. The DeLong test (DeLong et al., 1988) is a nonparametric test that can be used to compare differences between two AUROC scores, based on the theory of generalized $U$-statistics. It is also possible to use bootstrap replicates of the data to make this comparison, as available in the R function `roc.test`. Everingham et al. (2015) made use of the bootstrap to compare AP scores.

**Q2:** This situation seems to be considered much more rarely in machine learning, but notably the DELVE team did address this question, when using multiple independent training sets and a common test set. Rasmussen (1996a, sec. 2.5) used a 2-way ANOVA model for comparing the difference in losses between methods A and B, with random effects for both the choice of training set and the choice of test example. The goal was to assess if the mean difference is significantly different from zero. This was carried out using a quasi-F test.

Benavoli et al. (2017) propose a correlated $t$-test when a resampling procedure such as cross-validation is used, so that the training sets are no longer independent. As noted above, they used comparisons of the *average* performance on each test fold, and not on each individual test case.

**Q3:** Demšar (2006) focuses on the the statistical comparison of classifiers over multiple tasks. He discusses a repeated-measures ANOVA analysis, but recommends instead the Friedman test, due to the strong assumptions necessary for ANOVA. The Friedman test assesses the ranked performance of each algorithm on each task, and uses a $\chi^2$ test to compare the observed ranks to the null hypothesis that all algorithms are equivalent.

If the null hypothesis is rejected with the Friedman test, then a post-hoc test (e.g., the Nemenyi test, Nemenyi 1963) can be used to compare pairs of predictors to assesses if the difference in their average ranks is statistically significant. This is similar to the standard ANOVA analysis, where if the null hypothesis that the predictor effect is zero is rejected, "post-hoc" tests are carried out pairwise—see, e.g., Montgomery (2013, sec. 3.5) for further details.

**Effect sizes:** One of the criticisms of NHST is that having enough data can often reject the null hypothesis for arbitrarily small effects (see discussion in sec. 5.2). This can be seen, for example, in the the $t$-test statistic $t = \frac{m}{s/\sqrt{N}}$, when increasing $N$ makes the statistic larger. To guard against this, it is often recommended that

not only a $p$-value but also an effect size be reported, see, e.g., Appelbaum et al. (2018). The book by Cohen (1988, p. 9) defines the effect size as "the degree to which the phenomenon is present in a population". The effect size can be based on properties such as group differences, risk estimates, and indices of association. Below we discuss two examples, Cohen's $d$ and Cohen's $g$, which are used in the worked examples in sec. 6.

Cohen's $d$ is defined as the difference between two means divided by a standard deviation for the data, $d = m/s$ in the notation used above. Note that, in comparison with the $t$-statistic, there is no $\sqrt{N}$ present. For calibration Cohen (1988, p. 40) gives rules of thumb that a value of $d = 0.2$ can be described as "small", 0.5 as "medium" and 0.8 as "large".

For 0/1 losses, one can compute $\hat{\phi} = n_{01}/(n_{01} + n_{10})$, the maximum likelihood estimator of the fraction of (01) cases relative to the total of (01) and (10) cases. Under the null hypothesis this fraction would be 1/2, so the difference is $g = \hat{\phi} - 1/2$, which is known as Cohen's $g$ (Cohen, 1988, pp. 147-149). His rules of thumb are that values of $|g| < 0.05$ are regarded as "negligible", in the interval $(0.05, 0.15)$ as "small", $(0.15, 0.25)$ as medium and $|g| > 0.25$ as "large".

### 5.2 Criticisms of point null hypothesis significance testing

**Frequentist analysis:** Here we define the null hypothesis $H_0 : \Delta = 0$, and an alternative hypothesis (e.g. $\Delta \neq 0$). For null hypothesis significance testing (NHST) we then select a statistical test and compute the relevant test statistic (e.g., the $t$ statistic given in sec. 5.1 for the paired $t$-test). To assess the significance of the observed test statistic, we derive its distribution under $H_0$. One then sets a significance level $\alpha$ (e.g., 5% or 1%) corresponding to the maximum acceptable false positive rate for the test, and derives the rejection region of the test, i.e. values of the statistic for which $H_0$ will be rejected. Depending on the observed value of the statistic and the chosen $\alpha$, the decision is either to reject the null hypothesis, or to not reject it. Alternatively one can calculate the value of $\alpha$ which would just lead to rejection of $H_0$, and report it as the $p$-value for the test. See, e.g., Wasserman (2004, ch. 12) for more discussion of hypothesis testing and $p$-values.

There has been much criticism of $p$-values in recent years, see e.g., the 2016 statement by the American Statistical Association (ASA) (Wasserstein & Lazar, 2016). See also sec. 5.5.4 in Murphy (2022) entitled "p-values considered harmful". Below we reproduce three of the six principles from the ASA Statement Wasserstein & Lazar (2016), along with a comment.

Principle 2: **The p-value is not the probability that the null hypothesis is true.** *Researchers often wish to turn a p-value into a statement about the truth of a null hypothesis, or about the probability that random chance produced the observed data. The p-value is neither. It is a statement about data in relation to a specified hypothetical explanation, and is not a statement about the explanation itself.*

Principle 3: **Scientific conclusions and business or policy decisions should not be based only on whether a p-value passes a specific threshold.**

Principle 5: **A p-value, or statistical significance, does not indicate the size an observed effect, or the importance of a result.**
Comment: The $p$-value does not separate between the effect size and the sample size—having enough data can often reject the null hypothesis for arbitrarily small effects.

**Bayesian analysis:** For a Bayesian analysis we combine a prior $p(\Delta)$ with a likelihood model $p(\mathbf{d}|\Delta)$ to obtain a posterior distribution $p(\Delta|\mathbf{d})$ . (This may require dealing with some additional parameters—see, for example, the noise variance $\sigma^2$ in the Bayesian $t$-test described in sec. 5.4.)

One can also carry out hypothesis testing under the Bayesian framework, when considering the point null hypothesis $H_0 : \Delta = 0$ (a delta-function prior for $p(\Delta)$) against the alternative $H_1$ which uses a broad prior

$p(\Delta)$; see, e.g., Kruschke (2015a, sec. 12.2). We then have

$$p(\mathbf{d}|H_i) = \int p(\mathbf{d}|\Delta, \tau)p(\Delta, \tau|H_i) \, d\Delta d\tau \qquad i = 0, 1 \tag{4}$$

where $\tau$ denotes the additional (or nuisance) parameters. $p(\mathbf{d}|H_i)$ is called the marginal likelihood. But note that the marginal likelihood is very sensitive to the specification of the prior, and the use of noninformative priors is dangerous in this context; see e.g., Kass & Raftery (1995, sec. 5.1) who refer to "Bartlett's paradox" (Bartlett, 1957).

To compute the posterior probability $p(H_i|\mathbf{d})$ we have

$$p(H_i|\mathbf{d}) = \frac{p(\mathbf{d}|H_i)p(H_i)}{\sum_{j=0,1} p(\mathbf{d}|H_j)p(H_j)}, \tag{5}$$

where $p(H_0)$ and $p(H_1)$ are the prior probabilities of the two models, which may be taken as $1/2$. This general framework (without the point null constraint) is known as Bayesian model comparison.

This procedure is mathematically well-defined, but there has been much discussion about the use of the point null hypothesis $H_0$ in Bayesian hypothesis testing. For example Gelman et al. (2013, p. 95) state "In problems involving a continuous parameter $\theta$ (say the difference between two means), the hypothesis that $\theta$ is exactly zero is rarely reasonable". Benavoli et al. (2017, sec. 3) make the same point, and thus follow the parameter estimation plus equivalence testing approach (as discussed in the next section) rather than Bayesian model comparison. We agree with this viewpoint.

### 5.3 Equivalence testing

We are interested in comparing two predictors A and B, and can evaluate the loss on each test case and make paired comparisons. We wish to make inferences about a population parameter $\Delta$. In NHST one generally considers a point null hypothesis $\Delta = \Delta_0$. Alternatively one can consider a range of possible values around $\Delta_0$, i.e. the interval $(\Delta_0 - m_L, \Delta_0 + m_U)$ for lower and upper margins $m_L > 0$ and $m_U > 0$. The idea is that this interval indicates a "range of parameter values that are considered to be practically equivalent to the null value for purposes of the particular application" (Kruschke, 2015a, p. 336). Kruschke refers to this interval as the "region of practical equivalence" or ROPE for short[3]. This is the formulation for *equivalence testing* (see, e.g., the book by Wellek 2010). For the paired $t$-test the usual choice is $\Delta_0 = 0$ and a symmetric interval $m_L = m_U = m$.

Equivalence testing was initially developed for bioequivalence testing, where the issue is being able to state that a generic version of a drug is as effective as the primary manufacturer's formulation (Wellek, 2010, p. xii). But as (Wellek, 2010, p. 6) notes "problems of equivalence assessment are encountered in virtually every context where the application of the methodology of testing statistical hypotheses makes any sense at all. Accordingly, it is almost harder to identify a field of application of statistics where equivalence problems play no or at most a minor role, than to give reasons why they merit particular attention in some specific field." We believe that the equivalence testing framework has a large role to play in the comparison of machine learning methods.

**Specifying the ROPE:** If the quantity $\Delta$ is interpretable we should be able to set the ROPE based on our knowledge of the problem. For example the US Food and Drug Administration (FDA) guidance for drug bioequivalence is for an 80-125% range of efficacy of the new drug relative to reference drug (Wellek, 2010, p. 15).

In a machine learning classification problem, we may believe that a difference of 1% in classification accuracies is practically equivalent, and indeed this is the ROPE used in Benavoli et al. (2017). However, this choice

---

[3]Other names can be used of the ROPE, e.g., Schuirmann (1987) refers to it as the "equivalence interval"; other terms include the "range of equivalence" and "indifference zone", see Kruschke (2015a, p. 337)

may depend on the level of performance of the classifiers. For example if the accuracy is 95%, so that the error rate is 5%, an improvement of 1% would make a 20% reduction in the error rate to 4%. But if the error rate was 25%, a 1% change would be of much less importance. In general it is best to set the ROPE based on knowledge of the problem domain, and how (in the machine learning case) the predictions of the models will be used in practice.

If the parameter is less interpretable, a standard choice is to set the ROPE to $[\Delta_0 - \gamma s, \Delta_0 + \gamma s]$ where $s$ denotes the standard deviation of the difference vector, and $\gamma$ is a constant. One common choice is to set $\gamma = 0.1$, (Kruschke, 2015b; Makowski et al., 2025). This is based on the notion of the "effect size", defined as Cohen's $d$ (see sec. 5.1), with $d = 0.2$ taken to be a small effect. Other values of $\gamma$ such as 0.2 have also been used in the literature. Increasing the width of the ROPE will naturally lead to it being more likely that an inference of equivalent performance will be obtained. Table 1.1 in Wellek (2010) gives standard proposals for the limits for a variety of situations.

As an example of this construction, consider a Bernoulli variable with probability parameter $\phi$, so $s = \sqrt{\phi(1 - \phi)}$. This could arise, e.g., in estimating the fraction of (01) cases to the total of (01) and (10) cases. For $\phi = 0.5$ and $\gamma = 0.1$ we obtain a ROPE of $(0.45, 0.55)$, which is in agreement with a non-negligible value of Cohen's $g$ (see sec. 5.1). For $\phi = 0.9$ (or 0.1) we obtain a ROPE of $(0.47, 0.53)$. However, for an interpretable case like this it would be desirable to use knowledge of the problem and how the classifier will be applied to set the ROPE, if such information is available.

In sec. 6 we give some examples of how varying $\gamma$ affects the inferences that are drawn from given data.

**Bayesian analysis:** To use the ROPE in practice, we consider three regions in $\Delta$-space: (i) the ROPE, where the performance of A and B is regarded as practically equivalent; (ii) the region outside of the ROPE where A practically outperforms B; and (iii) the region outside of the ROPE where B practically outperforms A. Let us denote these regions as $(A \equiv B)$, $(A > B)$ and $(A < B)$.

Making use of the posterior over $\Delta$ we can compute the probability masses $p(A \equiv B)$, $p(A > B)$, $p(A < B)$. To draw inferences, suppose we define a threshold $\theta$ such as 0.95 (although other values like 0.90 or 0.99 could also be used). Then the decision rule is that if a fraction $\theta$ or more of the probability mass lies within the ROPE, we decide that A and B are practically equivalent; if a fraction $\theta$ or more of the probability mass lies within either of the regions $A > B$ or $A < B$ we make that decision; otherwise we cannot decide if the methods are practically equivalent or practically different. Clearly making the ROPE wider will place more probability mass in $p(A \equiv B)$, and less in the other two regions.

The above procedure is termed "ROPE (full)" by Makowski et al. (2019). An alternative approach recommended by Kruschke (2015a, sec. 12.1) is to calculate the highest density interval (HDI) for the posterior; this is the span of values that are most credible and contain $1 - \alpha$ of the mass of the distribution. This credible interval is a Bayesian analogue to a frequentist confidence interval. Give the HDI and ROPE, decision rule is as below (Kruschke, 2015a, sec. 12.1): (i) A parameter value is declared to be not credible, or rejected, if its entire ROPE lies outside the $1 - \alpha$ HDI of the posterior distribution of that parameter; (ii) A parameter value is declared to be accepted for practical purposes if that value's ROPE completely contains the $1 - \alpha$ HDI of the posterior of that parameter; (iii) When the HDI and ROPE overlap, with the ROPE not completely containing the HDI, then neither of the above decision rules is satisfied, and we withhold a decision. Makowski et al. (2019) term the above procedure HDI-ROPE, but their recommendation is to prefer ROPE-(full).

With Bayesian methods there is always the question of sensitivity to the model assumptions. But note that inference for $\Delta$ on a single task is a low-dimensional problem (it may be more than one-dimensional due to "nuisance" parameters), and that it is typically more difficult to understand the effect of priors in high-dimensional situations. For the Bayesian $t$-test and Bayesian McNemar tests described below one can use standard non-informative prior constructions. In these parameter estimation cases the posterior will be dominated by the likelihood term rather than the prior as $N$ gets large. But there is scope to investigate sensitivity to model assumptions; for example, as discussed in sec. 5.4, a Gaussian model for $p(d_j|\Delta)$ can be replaced with a more robust Student-$t$ model if evidence suggests there is a poor fit of the Gaussian model to the data.

Bayesian hierarchical models for the multiple task setting will inevitably be more complex, involving inference for a performance parameter (e.g., a $\Delta_i$) for each task $i$, and an overall performance parameter $\Delta_0$ for the set of tasks, as discussed in sec. 5.6. However, the parameters involved are still interpretable, aiding exploration of sensitivity to modelling assumptions.

**Frequentist analysis:**  There are a number of frequentist testing setups that make use of the ROPE (see, e.g., Lakens, Scheel, and Isager 2018). In the *minimal-effects test* the null hypothesis is that $\Delta$ lies within the ROPE, and the alternative hypothesis is that it lies either to the left or right of the ROPE. The question posed is then is whether the null hypothesis can be rejected. In the *equivalence test* the roles of the null and alternative hypotheses above are reversed. There is also a *superiority test* with a null hypothesis that $\Delta < \Delta_0 + m_U$, against the alternative that $\Delta > \Delta_0 + m_U$. There is a corresponding *inferiority test*, and also a a *non-inferiority test* with a null hypothesis that $\Delta < \Delta_0 - m_L$, against the alternative that $\Delta > \Delta_0 - m_L$. An example of non-inferiority testing is to show that a generic drug is not appreciably worse than an existing treatment. Fig. 1 in Lakens et al. (2018) nicely illustrates the various setups.

A classic approach to equivalence testing is the "two one-sided tests" (TOST) procedure (Schuirmann, 1987). This carries out two tests, (i) $\Delta \leq \Delta_0 - m_L$ and (ii) $\Delta \geq \Delta_0 + m_U$. Let the $p$-values for these two tests be denoted by $p_L$ and $p_U$ respectively. Under the TOST procedure a conclusion of statistical equivalence can be made at level $\alpha = \max(p_L, p_U)$. One can also think of the TOST procedure in terms of a confidence interval (see, e.g., Schuirmann 1987). The null hypothesis for equivalence testing (i.e. that $\Delta$ lies to the left or right of the ROPE) is rejected at significance level $\alpha$ if the $(1 - 2\alpha)$ confidence interval fully lies within the ROPE. In fact the TOST procedure is not the optimal equivalence test, as discussed by Romano (2005) and Möllenhoff et al. (2022) (cited in Campbell and Gustafson 2024).

For paired data, chapter 5 in Wellek (2010) gives a paired $t$-test for equivalence, and an equivalence test for the McNemar setting.

**Comparison of Bayesian and frequentist equivalence testing:**  We start by noting the general point that Bayesian and frequentist analyses are addressing different setups. In the Bayesian view, parameters are random variables with distributions, the data are fixed, and inferences are based on the posterior distribution of parameters. In the frequentist approach, parameters are fixed (but unknown), the data are treated as random, and inferences are based on the sampling distribution of the data. Despite this, the two approaches can sometimes lead to similar inferences. For example, Jaynes (1976) showed that for a single unknown location parameter having a uniform prior, the (Bayesian) credible interval and the confidence interval coincide. He also showed a similar result for a single scale parameter under a Jeffreys' prior.

One attraction of the Bayesian ROPE-(full) approach is that it returns probabilities for the three regions, i.e. $p(A \equiv B)$, $p(A > B)$, $p(A < B)$, allowing us to assess the merits of various hypotheses. In contrast, frequentist testing is set up with various (related) null hypotheses for minimal-effects, equivalence, superiority testing etc. If we care specifically about (say) equivalence testing for bioequivalence this is reasonable, but otherwise it is rather constraining.

Linde et al. (2023) have carried out a simulation study on decisions about equivalence made by the TOST, HDI-ROPE and ROPE-(full) approaches. In fact Linde et al. (2023) describe their third method as "Bayes Factor", but this is very similar to the ROPE-(full) approach. To see this let $H_1$ to indicate the ROPE region, $H_0$ denote its complement, and $\mathbf{d}$ the paired differences. (The ROPE region is denoted as $H_1$, because the null hypothesis $H_0$ for frequentist equivalence testing is that the parameter lies outside the ROPE.) We have that

$$\frac{p(H_1|\mathbf{d})}{p(H_0|\mathbf{d})} = \frac{p(\mathbf{d}|H_1)}{p(\mathbf{d}|H_0)} \cdot \frac{p(H_1)}{p(H_0)}. \tag{6}$$

The first term on the RHS is the Bayes factor, and the second is the prior odds. Here $p(H_1)$ is the mass of the prior $p(\Delta)$ in the ROPE region, and $p(H_0) = 1 - p(H_1)$; this is the nonoverlapping hypotheses (NOH)

setup of Morey & Rouder (2011). So we see that inferences for the ROPE-(full) and Bayes Factor approaches are closely related.

Linde et al. (2023) state that

> The results indicate that the Bayes factor interval null approach compares favorably to the other two approaches in terms of statistical power. Critically, compared with the Bayes factor interval null procedure, the two one-sided tests and the highest density interval region of practical equivalence procedures have limited discrimination capabilities when the sample size is relatively small. [...] Because of these results, we recommend that researchers rely more on the Bayes factor interval null approach for quantifying evidence for equivalence, especially for studies that are constrained on sample size.

Also note that the TOST and HDI-ROPE procedures gave similar results to each other. It is worth noting that for larger sample sizes and margins the results of the three different methods can be similar.

There was some criticism of Linde et al.'s study by Campbell & Gustafson (2024), who showed that the performance of the HDI-ROPE, Bayes factor and the optimal frequentist test (OT) approaches can often well-approximate one another when they are calibrated to have the same Type I error rate. However, Linde et al. (2024) argue that this calibration requires "the use of extreme, highly unorthodox settings" (e.g., thresholds, significance levels etc.) and consequently they still argue for the practical advantages of the Bayes Factor approach.

Campbell & Gustafson (2024) also give a useful discussion of optimality under the frequentist and Bayesian approaches. In the frequentist approach, one aims to minimize Type II errors subject to an upper bound on the maximum probability of a Type I error. In contrast, if the parameters are drawn from the overall prior distribution (a mixture of $H_0$ and $H_1$) then the Bayes factor procedure minimizes the average loss (the Bayes' risk) (Berger, 1985).

One other advantage of Bayesian approaches is when comparing methods over multiple tasks (Q3); in this case one can use hierarchical Bayesian models, as discussed below in sec. 5.6. We are not aware of any frequentist methods to tackle equivalence testing for this problem. Wellek 2010, ch. 7 does address "multisample tests for equivalence" but these are asking a different question, as to whether the performance of several different methods are all equivalent.

Given the findings of Linde et al. (2023) and Linde et al. (2024), (especially for small sample sizes), we recommend the ROPE-(full) procedure for the comparison of methods, and use it below.

### 5.4 Bayesian $t$-test for difference in means

Consider Q1, the comparison of two different predictors A and B for a given task on the basis of a common test set. Let $z_{ij}$ be the loss of predictor $i$ on test case $j$. Now consider the *paired difference* $d_j = z_{Aj} - z_{Bj}$, and let $\mathbf{d} = (d_1, \ldots, d_N)$. As discussed around eq. 2, it makes sense to consider the *paired* difference, as a random effect that makes a given example hard to predict (e.g., larger variability in the ground-truth $p(y|x)$) will be common across both predictors. See Appendix A for more on this point.

We first assume that $d_j \sim N(\Delta, \sigma^2)$, where $\Delta$ is the mean difference in losses, and $\sigma^2$ is the variance of the distribution. We also assume that the $d_j$'s are sampled independently. We are interested in the posterior $p(\Delta|\mathbf{d})$. Using an non-informative prior over the parameters $(\Delta, \sigma^2)$, Murphy (2012, sec. 4.6) shows that

$$p(\Delta|\mathbf{d}) = t_{N-1}(\Delta|m, s^2/N), \tag{7}$$

where $m = \frac{1}{N}\sum_{j=1}^{N} d_j$, the sample mean, $s^2 = \frac{1}{N-1}\sum_{j=1}^{N}(d_j - m)^2$ is an unbiased estimate of the variance, and $t_{N-1}$ is the Student-$t$ distribution with $N - 1$ degrees of freedom.

The derivation of eq. 7 above makes use of a non-informative prior over the parameters $(\Delta, \sigma^2)$. Alternatives have been proposed; for example Morey & Rouder (2011) consider a proper Student-$t_\nu$ prior on the effect size, combined with a improper prior on $\sigma^2$. A special case when $\nu = 1$ is the Cauchy prior.

The above model makes strong assumptions, e.g. about the normality of the differences. The standard model checking approach to assess this is to make a quantile-quantile (Q-Q) plot of the differences data. One way to handle violations of normality is to robustify the model for $p(d_i|\Delta)$; for example following Kruschke (2015a, sec. 16.2) one might use a Student-$t$ distribution rather than a normal distribution. This is less amenable to an analytical treatment, but is easy to handle with MCMC methods, such as the `brms` library in R (Bürkner, 2017). A non-parametric frequentist alternative to the $t$-test is the Wilcoxon signed-rank test, which assumes that the observations are drawn from a symmetric distribution and are i.i.d. Note that Benavoli et al. (2017, sec. 4.2.1) developed a Bayesian signed-rank test, based on a Dirichlet process prior.

This paired testing model can also be used for Q2, the comparison of two machine learning algorithms, taking into account the fixed factor of different algorithms, and the random factor being the choice of independent training sets. Here we have assumed that the test set is large, so that one can use the average loss over the test set as $z_{ij}$.

Benavoli et al. (2017) also address the effect of training set variability, by carrying out repeated runs of a $k$-fold cross validation procedure. Here the difference of accuracies is computed on each test fold (note: *not* on each test case). They handled the lack of independence between training sets by developing a correlated $t$-test.

### 5.5  Bayesian McNemar test

The paired $t$-test cannot be applied to the comparison of two binary classifiers that make "hard" 0/1 predictions—in this case there are only three possible values of the difference (1, 0 and -1), and it is not a credible assumption that these are drawn from a normal distribution.

We have seen in sec. 5.1 that the (frequentist) McNemar's test can be used in this situation. Chechile (2020, sec. 4) discusses a "Bayesian McNemar's test" that can be used for the analysis of paired categorical responses. Recall that the counts associated with the four different outcomes are denoted $n_{00}$, $n_{01}$, $n_{10}$, and $n_{11}$ respectively, with their sum being $N$, the number of test cases. Let the probabilities of the four outcomes be $\theta_{00}, \theta_{01}, \theta_{10}, \theta_{11}$. Under the null hypothesis that the error rates are equal, we have that $\theta_{00}+\theta_{01} = \theta_{00}+\theta_{10}$, or that $\theta_{01} = \theta_{10}$.

An appropriate Bayesian model for this 4-outcome case is the Dirichlet distribution. Given prior parameters $\alpha_{00}, \alpha_{01}, \alpha_{10}, \alpha_{11}$ for the four cells, the posterior after observing the counts is again Dirichlet with parameters $\alpha_{ij} + n_{ij}$ for $i, j \in \{0,1\}$. A common non-informative choice for the prior parameters is to set $\alpha_{ij} = 1$ for $i, j \in \{0,1\}$. However, we wish to focus attention on the (01) and (10) cells. To draw a sample from the Dirichlet distribution one can make use of a "stick breaking" construction that samples from marginal and conditional Beta distributions, see e.g. Chechile (2020, Theorem 3.5). This can be used to show that the fraction $\phi = \theta_{01}/(\theta_{10} + \theta_{01})$ after observing the data is distributed as $\text{Beta}(\alpha_{01} + n_{01}, \alpha_{10} + n_{10})$, see Chechile (2020, sec. 4.3.2). Note that as the (01) event denotes *misclassifications* by algorithm A, values of $\phi$ which are less than 0.5 indicate that A outperforms B. Lacoste et al. (2012, sec. 4) provide a similar argument to Chechile (2020) by direct integration of the Dirichlet distribution. They effectively compute $p(\text{left}) = \int_0^{0.5} p(\phi|n_{01}, n_{10}, \alpha_{01}, \alpha_{10})d\phi$ and declare algorithm A to be superior if $p(\text{left}) > 1/2$, and algorithm B otherwise.

To define a ROPE for this case, consider the variance of a 0/1 Bernoulli random variable $W$ drawn with probability $\phi$. We have that the mean of $\phi$ is $\bar{\phi} = \alpha_{01} + n_{01}/(\alpha_{10} + n_{10} + \alpha_{01} + n_{01})$. Then $\text{var}(W) = \mathbb{E}[(W - \bar{\phi})^2] = p(W = 1)(1 - \bar{\phi})^2 + p(W = 0)\bar{\phi}^2 = \bar{\phi}(1 - \bar{\phi})^2 + (1 - \bar{\phi})\bar{\phi}^2 = \bar{\phi}(1 - \bar{\phi})$. Thus the standard deviation is $s = \sqrt{\bar{\phi}(1 - \bar{\phi})}$, and one can use the usual ROPE construction of $[0.5 - \gamma s, 0.5 + \gamma s]$.

### 5.6  Multiple tasks: Bayesian hierarchical models

In this section we consider Q3, the comparison of two predictors across *multiple* tasks. The approach we shall follow is to use a Bayesian *hierarchical* model. These are discussed, for example, in chapter 5 of Gelman et al. (2013). The use of such hierarchical models assumes that the losses across the different tasks are commensurate.

**Bayesian hierarchical $t$-test:** Let $\mathbf{d}_i$ be the vector of loss differences for task $i$ over the test cases for that task. For a given task we consider a model as in sec. 5.4, so that for test case $j$ of task $i$, we may assume that $d_{ij} \sim N(\Delta_i, \sigma_i^2)$. Following Benavoli et al. (2017, sec. 4.3.1) we now consider a *hierarchical* model so that for $q$ tasks we have $\Delta_1, \ldots, \Delta_q \sim t_\nu(\Delta_0, \sigma_0^2)$ and $\sigma_1, \ldots, \sigma_q \sim U(0, \bar{\sigma})$. Here the $\Delta_i$'s are drawn from a $t$-distribution with $\nu$ degrees of freedom, with mean $\Delta_0$ and scale parameter $\sigma_0$; the $\sigma_i$s are drawn from a Uniform distribution on $[0, \bar{\sigma}]$. BCDZ set $\bar{\sigma}$ to be a large multiple of $\bar{s}$, the average of the empirical standard deviations of the $\mathbf{d}_i$ vectors across the $q$ tasks. They argue that the inferences are insensitive to $\bar{\sigma}$ as long as it is set large enough. The model is completed with uniform priors for $\Delta_0$ and $\sigma_0$ and a prior on $\nu$. Obviously one might choose a somewhat different parameterization for various parts of the hierarchical model, but our description here follows BCDZ.

The most important parameter in the hierarchical model is $\Delta_0$, which is the average difference in loss across the datasets. We therefore wish to explore $p(\Delta_0|D, \psi)$, where $D = (\mathbf{d}_1, \ldots, \mathbf{d}_q)$ and $\psi$ denotes the various parameters in the prior distributions. Computations for the hierarchical model can be obtained via Markov chain Monte Carlo (MCMC) sampling.

A simpler alternative to the hierarchical model would be to compute the mean difference on each dataset, and then do a (Bayesian) $t$-test on this vector of these differences. But this ignores the variability within the different datasets, which might differ across datasets—this is modelled explicitly in the hierarchical model.

**Bayesian hierarchical McNemar test:** The Bayesian McNemar test (see sec. 5.5) can also be extended to a hierarchical model if there are multiple tasks, each with $0/1$ loss data. In this case there would be $q$ parameters $\phi_1, \ldots, \phi_q$, being the true proportion for cell (01) against cells (10) and (01) in each task. A hierarchical model would assume that these parameters are drawn from a common prior. One possibility is to make this a Beta$(a, b)$ distribution, with $a$ and $b$ drawn from a suitable hyperprior. For example Gelman et al. (2013, sec. 5.3) discuss reparameterizing this in terms of priors on $\rho_1 = \log(a/b) = \mathrm{logit}(a/a + b)$ and $\rho_2 = \log(a + b)$. Choosing a diffuse hyperprior which is uniform on $(\frac{a}{a+b}, (a+b)^{-1/2})$ results in a hyperprior in $(a, b)$ space of $p(a, b) \propto (a+b)^{-5/2}$.

Given the count data for each of the $q$ tasks, samples from the posterior distribution for $a$ and $b$ can be obtained, In our experiments below we have used the `hef` function in the `bang` (Bayesian Analysis, No Gibbs) library in R by Northrop & Hall (2025) to carry out this sampling using the generalized ratio-of-uniforms method.

The aim of the hierarchical model is to predict $\phi_{next}$, the $\phi$ parameter for the next dataset drawn from the same underlying distribution as the $q$ datasets given the observed multitask data $D$. We have that

$$p(\phi_{next}|D) = \int p(\phi_{next}|a, b)p(a, b|D) \, da \, db \simeq \frac{1}{S} \sum_{s=1}^{S} p(\phi_{next}|a_s, b_s) \tag{8}$$

where the samples $\{a_s, b_s\}_{s=1}^{S}$ are drawn from the posterior $p(a, b|D)$. We have that

$$\mathbb{E}[\phi_{next}|D] \stackrel{def}{=} \bar{\phi}_{next} \simeq \frac{1}{S} \sum_{s=1}^{S} a_s/(a_s + b_s), \tag{9}$$

where we have used that the mean of a beta distribution with parameters $a_s, b_s$ is $a_s/(a_s + b_s)$. We can also approximate the distribution $p(\phi_{next}|D)$ as per eq. 8.

**Comparison of incommensurate datasets:** To build a hierarchical model it is necessary to assume that the parameters $\Delta_1, \ldots, \Delta_q$ or $\phi_1, \ldots, \phi_q$ are drawn from a common prior. If the losses are *incommensurate*, this is not reasonable. In this case Lacoste et al. (2012) pose the question "Does algorithm A have a higher chance of producing a better classifier than algorithm B in the given context?". This question is refined for a context $\mathcal{W}$ by defining

$$\bar{q}_{AB|\mathcal{W}} \stackrel{def}{=} \mathbb{E}_{(\mathcal{D}, n) \sim \mathcal{W}} \, \mathbb{E}_{S \sim \mathcal{D}^n} \, I[A(S) \stackrel{\mathcal{D}}{\succ} B(S)], \tag{10}$$

and declaring that algorithm A outperforms algorithm B in context $\mathcal{W}$ (denoted $A \stackrel{\mathcal{W}}{\succ} B$) if $\bar{q}_{AB|\mathcal{W}} > 1/2$. In eq. 10 the outer expectation is over tasks denoted by $\mathcal{D}$ drawn from the context $\mathcal{W}$, and the inner one

is over training sets $S$ of size $n$ drawn from task $\mathcal{D}$. The notation $A(S) \overset{\mathcal{D}}{\succ} B(S)$ means that algorithm A using training set $S$ outperforms algorithm B on the same training set, i.e., that the expected loss (or risk) of algorithm A is less than algorithm B. $I[a]$ denotes the indicator function, with $I[a] = 1$ if $a$ is true, and 0 otherwise. Lacoste et al. (2012, sec. 5) develop a Poisson Binomial test to estimate $\bar{q}_{AB|\mathcal{W}}$ in the case of 0/1 losses.

The above analysis simply accumulates the fraction of "wins" for algorithm A against B over tasks and associated datasets. However, when using a ROPE there are a number of options that could be employed. One suggestion is to set a threshold $\theta$ and count the number of wins for the four outcomes $A > B$, $A < B$, $A \equiv B$, and when the methods are neither practically equivalent or practically different.

## 5.7 Comparing multiple predictors

Above we have argued that it usually makes sense to choose a reference predictor, and to compare the other predictors to it. In the NHST framework one has to handle such *multiple comparisons* with care. One common approach is the Bonferroni correction, see e.g., Wasserman (2004, sec. 10.7). Consider a single test with type I error $\alpha^*$, i.e. the probability of erroneously rejecting a true null hypothesis is $\alpha^*$. Then if we conduct a family of $m$ such tests independently, the family-wise error rate (FWER, the probability of making at least one type I error in the family) is $1 - (1 - \alpha^*)^m$. We seek to have FWER $\leq \alpha$. The Bonferroni correction achieves this by setting $\alpha^* = \alpha/m$.[4] So if we were, for example, conducting $m$ paired $t$-tests, the FWER can be controlled by using a significance level of $\alpha/m$ for each one.

The above discussion relates to the issue of multiple comparisons with NHST, and controlling type I errors. But how is this issue addressed under a Bayesian analysis? In their paper entitled *Why We (Usually) Don't Have to Worry About Multiple Comparisons*, Gelman et al. (2012) state that "we are typically not terribly concerned with Type 1 error because we rarely believe that it is possible for the null hypothesis to be strictly true." I.e., by modelling of the situation with a posterior on $\Delta$, and the use of the ROPE to avoid a point null hypothesis, the issue of multiple comparisons is mitigated.

We have recommended making comparisons between a reference algorithm and each of the alternative algorithms in a pairwise fashion. It would be possible to consider a hierarchical model over algorithms (as opposed to tasks, as in sec. 5.6), with a prior on the "algorithm effects" (the $\alpha_i$'s in the ANOVA model of eq. 1). This would lead to joint inferences for these effects, which could then be used to consider the differences, e.g. $\alpha_i - \alpha_{i'}$. Although this approach has merits, we believe it is rather simpler to focus on the difference vector for comparing two algorithms, and leave exploration of this issue to future work.

## 5.8 Software

The code made available by Benavoli et al. (2017) in Python and R at `https://github.com/BayesianTestsML/tutorial` can be used for the Bayesian $t$-test and for the hierarchical model discussed in sec. 5.6. This repository also contains code for several other Bayesian nonparametric tests.

R code for the Bayesian McNemar test and for the hierarchical extension is available from our repository [link removed for double-blind review]. The latter makes use of the hierarchical exponential family (`hef`) function from Northrop & Hall (2025). Our repository also contains code for the worked examples in sec. 6.

There are many software resources for Bayesian modelling in general, including various packages for R, and libraries in Python (such as PyMC[5]) Also Stan[6] enables sophisticated Bayesian statistical modelling, and has interfaces for Python, R and Julia.

---

[4]The validity of the Bonferroni correction does not in fact require independence of the tests; it can be proved simply by using Boole's inequality, see e.g., Goeman & Solari (2014).

[5]PyMC: `https://www.pymc.io`.

[6]Stan: `https://mc-stan.org`.

# 6 Worked examples

We provide three worked examples. The first is for a binary classification task which is evaluated with the log loss using the Bayesian $t$-test. The second example is the MNIST digits, a multiclass classification task. The is evaluated with the 0/1 loss and the log loss, and we also explore the effect of varying $\gamma$ in defining the ROPE for the Bayesian McNemar test. The third example compares two predictors on 11 related tasks. We use the Bayesian McNemar test on each task individually, and also the Bayesian hierarchical McNemar test for the 11-task context.

## 6.1 Comparing four classifiers on one task

We illustrate the above methods on a binary classification task taken from (Camilleri et al., 2024). Their general investigation concerns the analysis of patterns of behaviour of three mice in their homecage based on video data. As part of this study they need to detect whether each of three mice are observable at a given time. A given mouse is identified by an RFID tag, and it may not be visible due to occlusion, either by the other animals, or by materials in the cage (e.g., a tube).

The data was obtained from 200 two-minute snippets of video, with 100 snippets used for training, 40 for validation and 60 for testing. The original task was to classify if a given mouse was observable for each second of video based on features extracted from the RFID and video data. There are 20,343 examples in the test set. Note that this is an *imbalanced* task, as only 7% of the data were labelled "not observable".

Although there are 20,343 examples in the test set, these are derived from only 176 snippet-mouse combinations. The observations of a given mouse in a snippet are heavily autocorrelated, and are thus not independent. To handle this, we averaged the log probability scores for a given mouse in each snippet, and treated this as one observation, resulting in $N = 176$ in eq. 7.

We consider here four classifiers: logistic regression (LgR), Naïve Bayes using a multinomial encoding (NB), a multilayer perceptron (MLP) and a support vector machine (SVM) using the RBF kernel. The threshold was chosen to set the fraction of true observable examples misclassified as not observable for each to 8%, see Camilleri et al. (2024, sec. 5.1.2) for further details.

We are addressing Q1 here, i.e., the comparison of predictors on a single task. On the basis of results on the validation set, the LgR classifier was selected by Camilleri et al. (2024) for use in the full system. We thus compare the LgR model with the other three (generically denoted as X), and assess if the differences in results are significantly different. This is assessed by applying a Bayesian t-test to the differences in log probability[7] of the correct label under the LgR and X models,

**Bayesian t-test:** We set the ROPE to $[-0.1s, 0.1s]$ where $s$ denotes the standard deviation of the difference vector. This follows the advice given in Kruschke (2015b) and Makowski et al. (2025). Note that $s$ is the standard deviation of the data, not the standard error of the mean, so that it does not scale as $\sqrt{1/N}$.

As an example, consider the difference in the log scores of the LgR and SVM models. A histogram of these differences is shown in Fig. 1(a). Notice that the differences are generally greater than zero, reflecting that the log scores are generally higher for LgR than SVM. In fact the median difference is 0.51; the mean is 0.34, pulled down by the left-hand tail of the distribution, where there are a few examples where the SVM scores much better. Applying the Bayesian $t$-test with a Gaussian noise model,[8] we obtain Fig. 1(b). The posterior is quite tightly concentrated around the mean because of the $1/N$ scaling of the variance in eq. 7) with $N = 176$. The plot clearly shows that the probabilities associated with the three alternatives are $p(LgR < SVM) = 0.0, p(LgR \equiv SVM) = 0.0, p(LgR > SVM) = 1.0$. Thus with a threshold of $\theta = 0.95$ we can conclude that practically LgR outperforms SVM on the basis of the log scores. For comparison, the standard (frequentist) $t$-test rejects the null hypothesis with a $p$-value of $3.7 \times 10^{-11}$ ($t = 7.06$, $df = 175$). The effect size is given by Cohen's $d = 0.53$, which is termed a medium strength effect.

---

[7]For the SVM, the numerical output lying in $(0, 1)$ was treated as a probability.

[8]The function `correlatedBayesianTtest.R` from `https://github.com/BayesianTestsML/tutorial` was used.

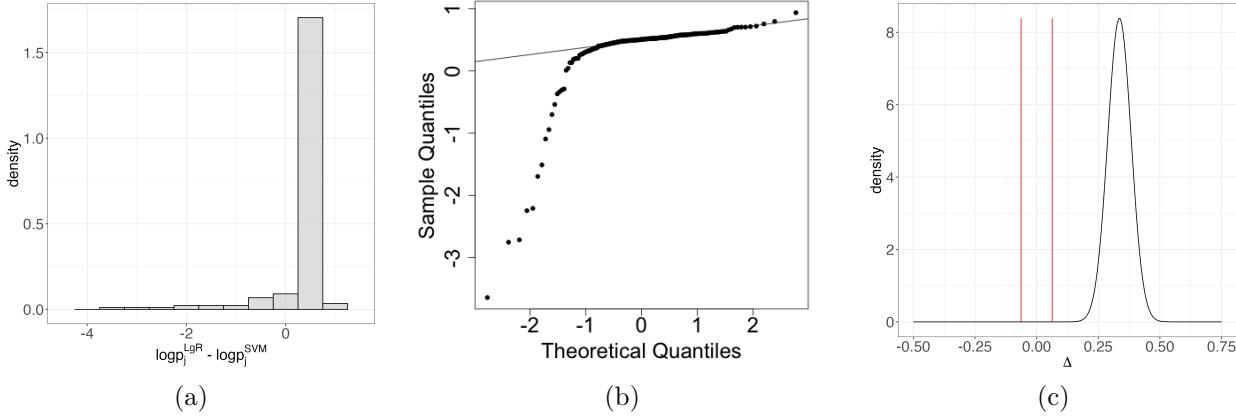

Figure 1: LgR vs. SVM. (a) Histogram of the differences $d_j = \log p_j^{LgR} - \log p_j^{SVM}$. (b) Q-Q plot of the $d_j$s. A line is drawn through the first and third quartiles. (c) The posterior distribution for $\Delta$ (black line), and the ROPE boundaries (shown in red).

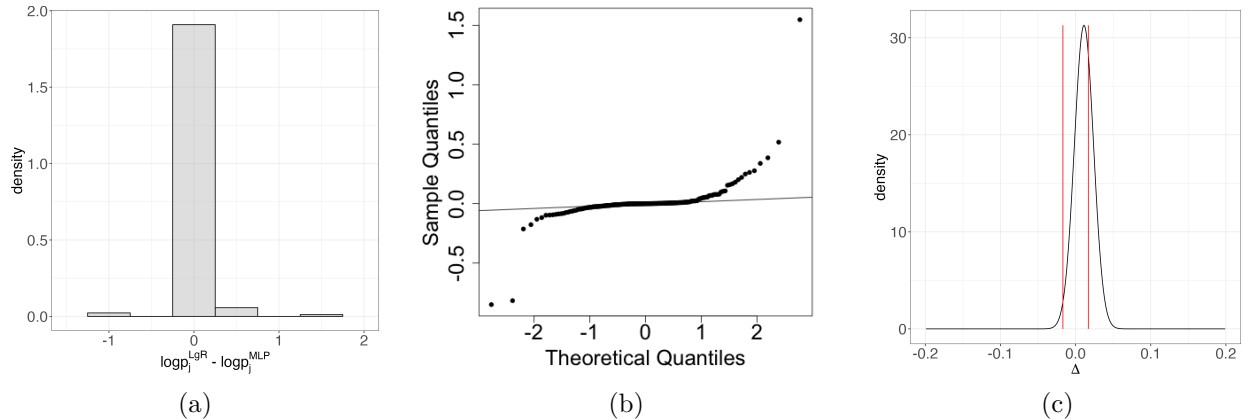

Figure 2: (a) LgR vs. MLP. Histogram of the differences $d_j = \log p_j^{LgR} - \log p_j^{MLP}$. (b) Q-Q plot of the $d_j$s. A line is drawn through the first and third quartiles. (c) The posterior distribution for $\Delta$ (black line), and the ROPE boundaries (shown in red).

The standard Bayesian $t$-test, as described in sec. 5.4, makes an assumption of normality for the $d_j$'s around $\Delta$. The histogram in Fig. 1(a) shows that the distribution is skewed around its median, and a Q-Q plot in Fig. 1(b) made with the R function `qqnorm` also shows a heavy tail (relative to the normal distribution) in the left hand tail. One way to handle this is to robustify the model for $p(d_j|\Delta)$ as discussed in sec. 5.4, by using a Student-$t$ distribution rather than the normal distribution. Doing so (and specifying priors for $\Delta$, the scale parameter $\sigma$ and degrees of freedom $\nu$) allows sampling for the posterior for $\Delta$ given the data. This was carried out with the `brms` R library (Bürkner, 2017). Using 3000 posterior samples, the ROPE probabilities were estimated as $p(LgR < SVM) = 0.0, p(LgR \equiv SVM) = 0.0, p(LgR > SVM) = 1.0$, which are in agreement with those obtained above.

For a contrasting example, consider the differences between the LgR and MLP classifiers, as shown in Fig. 2. Panel (a) shows the histogram of the differences in log scores. In this case the distribution appears to be symmetric about zero, and the mean and median are both very close to 0. The posterior for $\Delta$ (panel (c)) shows that the posterior mass spans all three regions, with $p(LgR < MLP) = 0.014, p(LgR \equiv MLP) = 0.660, p(LgR > MLP) = 0.326$. Thus with a threshold of $\theta = 0.95$ we cannot conclude that the two classifiers

are practically equivalent or practically different. For comparison, the standard (frequentist) $t$-test fails to reject the null hypothesis with a $p$-value of 0.38 ($t = 0.87$, $df = 175$), and the effect size is $d = 0.066$, which is regarded as "negligible".

Although the the distribution shown in Fig. 2(a) is much less skewed than Fig. 1(a), the Q-Q plot in Fig. 2(b) still suggests heavy tails. Using a similar robustified analysis to that described above gives rise to the result $p(LgR < MLP) = 0.018, p(LgR \equiv MLP) = 0.643, p(LgR > MLP) = 0.339$ on the basis of 3000 posterior samples, which is very similar to the results with a Gaussian noise distribution given above.

**Scaling the dataset size:** The above results show that for the LgR-SVM comparison, there is a very clear difference in performance, and that for the LgR-MLP comparison we cannot conclude that the two classifiers are practically equivalent or practically different. It is interesting to ask how the conclusions for the LgR-MLP comparison can change as we adjust the sample size, while fixing the sample mean $m$ and standard deviation $s$.

Multiplying the original sample size (176) by 6, we find that the $p$-value becomes 0.0324 (which would be significant at the $\alpha = 0.05$ level); it is as expected that the $p$-value will become significant as $N$ is increased. For this sample size the Bayes+ROPE probabilities are $p(LgR < MLP) = 0.000, p(LgR \equiv MLP) = 0.866, p(LgR > MLP) = 0.134$. The posterior for $\Delta$ tightens as $N$ is increased, and note from Fig. 2(c) that the MAP value lies within the ROPE region. Hence the probability mass in the ROPE region has increased relative to that for the original sample size, but still we cannot conclude that the two classifiers are practically equivalent or practically different.

Increasing the multiplier further to 15, we get a further decrease in the $p$-value to 0.0007, as expected. For the Bayes+ROPE analysis, the posterior probabilities now become $p(LgR < MLP) = 0.000, p(LgR \equiv MLP) = 0.960, p(LgR > MLP) = 0.040$. So at the $\theta = 0.95$ level, we can conclude that the methods are practically equivalent at this sample size.

**Calibrating the SVM output:** The above analysis illustrates significant differences between the LgR and SVM classifiers, but is in fact a bit misleading. It is well known that the outputs of a SVM are not well calibrated as probabilities. Platt (2000) showed that this can be mitigated by transforming the SVM scores with a sigmoid function, with adjustable offset and scaling parameters. Applying this "Platt scaling" to all of the classifiers, we find that for all three comparisons (MLP, NB and SVM against LgR), we cannot conclude that the two classifiers are practically equivalent or practically different.

## 6.2 Comparing two classifiers on MNIST

We illustrate the above methods on the famous MNIST 10-class classification task (LeCun et al., 1998), where the aim is to classify each example as belonging to one of the digits 0 through to 9. The dataset consists of 60,000 training examples and 10,000 test examples.

Two neural networks were trained for this task. The first was a multilayer perceptron (MLP) having one hidden layer with 800 hidden units, and a 10-class softmax output layer; this architecture was chosen based on the results given in Simard et al. (2003). The second was a convolutional neural network (CNN) with two layers of convolutional weights and max pooling, and a 10-class softmax output layer. The latter is taken from keras code examples,[9] and the MLP code was modified from that. For both architectures we averaged together the probabilistic predictions from 10 runs starting with different weight initializations. The MLP ensemble obtains a accuracy of 98.36% on the test set, with individual nets ranging from 98.09% to 98.33% performance. The CNN ensemble obtains a accuracy of 99.23%, with individual nets ranging from 99.09% to 99.24% performance.

**Bayesian McNemar test:** Below we consider hard classifications, based on the class with the highest predicted probability. The resulting counts are $n_{00} = 48$, $n_{01} = 29$, $n_{10} = 116$ and $n_{11} = 9807$. Given the

---

[9] https://keras.io/examples/vision/mnist_convnet/.

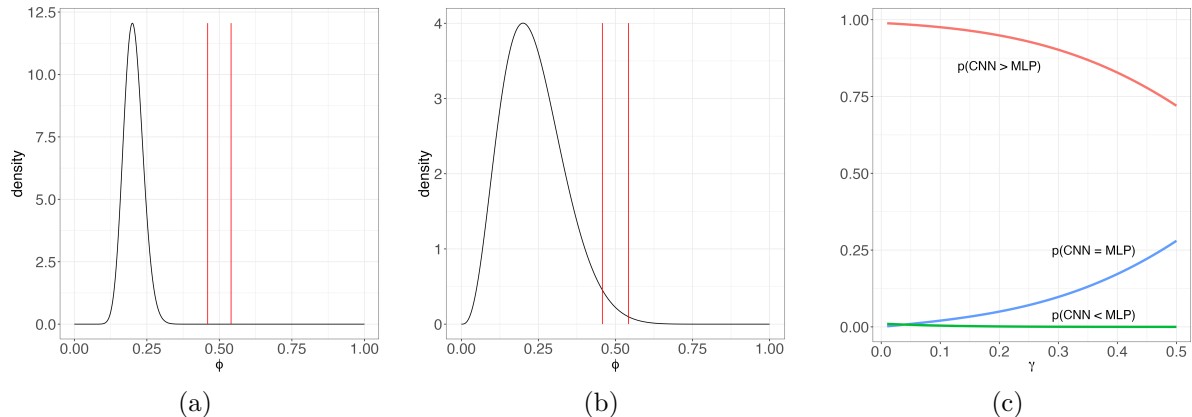

Figure 3: Bayesian McNemar test on MNIST. (a) and (b): The posterior distribution for $\phi$ (black line), and the ROPE boundaries (shown in red) for (a) $N = 10,000$ and (b) $N = 1,000$. (c) A plot of the three probabilities as a function of $\gamma$ for the $N = 1,000$ case.

high classification accuracies, it is not surprising that $n_{11}$ is the dominant entry, meaning that for 9,807 out of the 10,000 examples both classifiers get them correct.

Fig. 3(a) shows the posterior for $\phi$. The posterior mean is $\bar{\phi} = 0.20$ and the corresponding standard deviation is $s = \sqrt{\bar{\phi}(1 - \bar{\phi})} = 0.40$. Setting $\gamma = 0.1$ this gives a ROPE of $[0.460, 0.540]$. The posterior probabilities for the three regions are $p(CNN > MLP) = 1.00$, $p(CNN \equiv MLP) = 0.00$, $p(CNN < MLP) = 0.00$. Hence we can conclude that CNN practically outperforms the MLP. The standard McNemar test gives a $p$-value of $9.2 \times 10^{-13}$ and would thus the result be judged as significant at the $\alpha = 0.05$ level. To estimate the effect size, we first compute $\hat{\phi} = n_{01}/(n_{01} + n_{10}) = 0.20$, and then Cohen's $g = \hat{\phi} - 0.5$. We have $|g| = 0.30$, which would be described as a "large" effect.

The above result is very clear cut, be we can make things more interesting by considering what would happen with only $N = 1,000$ test cases. The counts are now $n_{00} = 5$, $n_{01} = 3$, $n_{10} = 12$ and $n_{11} = 980$, and Fig. 3(b) shows the corresponding posterior for $\phi$. Unsurprisingly there is now much more uncertainty in the posterior for $\phi$, and we have $p(CNN > MLP) = 0.976$, $p(CNN \equiv MLP) = 0.020$, $p(CNN < MLP) = 0.004$. Hence we can still conclude that CNN practically outperforms the MLP at the $\theta = 0.95$ level. The standard McNemar test gives a $p$-value of 0.038, so the result would thus be judged as just significant at the $\alpha = 0.05$ level.

**Varying $\gamma$:** The results above were obtained with a ROPE of $[0.5 - \gamma s, 0.5 + \gamma s]$ with $\gamma = 0.1$. However, as mentioned in sec. 5.3 this choice is only a recommendation, and it is interesting to observe the effect of changing $\gamma$ on the posterior probabilities of the three regions. Fig. 3(c) shows a plot of $p(CNN > MLP)$, $p(CNN \equiv MLP)$ and $p(CNN < MLP)$ as a function of $\gamma$ in the range $[0.01, 0.5]$ for $N = 1,000$. As $\gamma$ increases the width of the ROPE increases, and thus $p(CNN \equiv MLP)$ increases monotonically with $\gamma$. This causes the other two probabilities to decrease as $\gamma$ increases. We find that $p(CNN > MLP) > 0.95$ for values of $\gamma$ up to 0.19.

**Log probability predictions:** Here we compute the log probability of the correct label under the models. With the log probability loss one can incur a very large penalty if the predicted probability for the correct label is close to 0. To guard against this we transformed the prediction $p_i$ for class $i$ into $p'_i = \delta + (1 - C\delta)p_i$, where $\delta$ is a small positive value and $C$ is the number of classes (10 for MNIST). This transformation maps

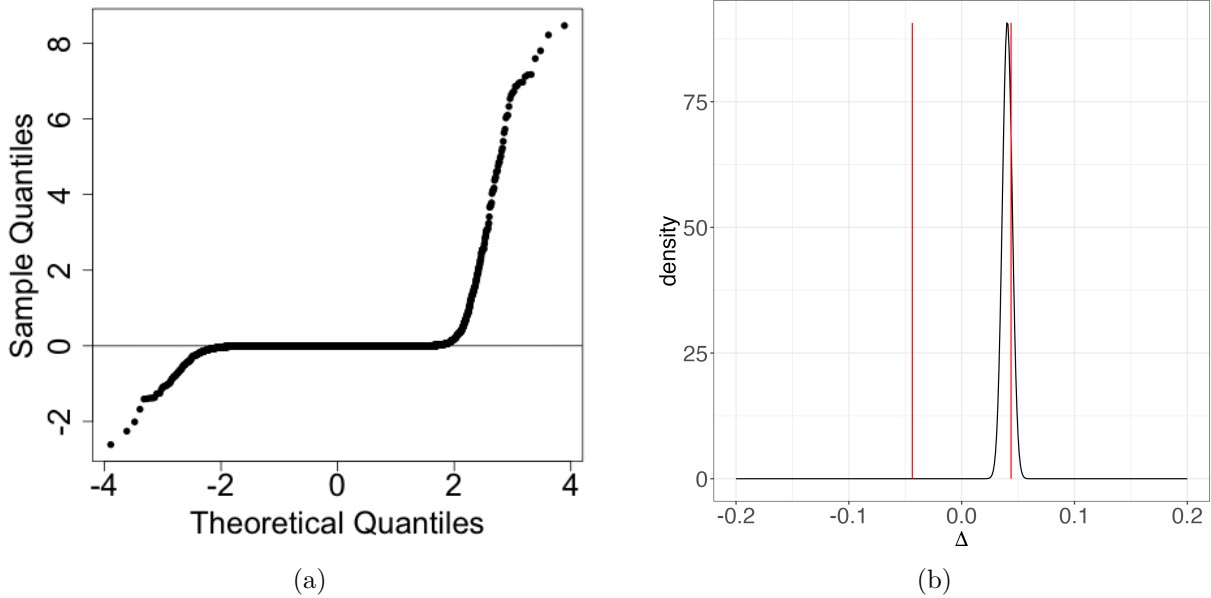

Figure 4: Log loss on MNIST. (a) Q-Q plot of $d_j = \log p_j^{CNN} - \log p_j^{MLP}$. A line is drawn through the first and third quartiles. (b) The posterior distribution for $\Delta$ (black line), and the ROPE boundaries (shown in red) for $N = 10,000$. under a Gaussian noise model.

each $p_i \in [0,1]$ to the range $[\delta, 1 - 9\delta]$ and ensures that $\sum_{i=0}^{9} p_i' = 1$. $\delta$ was optimized by a grid search for each network, giving values of 0.000001 for the CNN and 0.0002 for the MLP.[10]

We first visualize the data. Let $\log p_j^{CNN}$ denote the log predicted probability (or score) of the correct label for test case $j$ under the CNN model, and similarly for $\log p_j^{MLP}$. Both distributions have a concentrated peak just below 0 (indicating almost perfect predictions with probabilities near to 1), and a small number of examples in the left-hand tail, where the probabilities are lower. The minimum values for both log probabilities are below -8, which corresponds to a probability of the correct class of around 0.0003. By plotting the sorted values of $\log p_j^{CNN}$ and $\log p_j^{MLP}$ we can identify that about 300 of the 10,000 observations lie in the left-hand tail for both models.

Visualizing the paired log probability differences $d_j = \log p_j^{CNN} - \log p_j^{MLP}$ shows a large spike around 0, a left tail (with values below $-0.1$) of 134 examples, and a right tail (with values above 0.1) of 270 examples. The maximum difference is 8.47 and the minimum is $-2.61$. A Q-Q plot of the differences is shown in Fig. 4. The mean of the differences is 0.040, the median is 0.0018, and the standard deviation is 0.44.

Applying the Bayesian $t$-test with Gaussian noise and the ROPE set to $[-0.1s, 0.1s]$, we obtain Fig. 4(b), with probabilities $p(CNN < MLP) = 0.000$, $p(CNN \equiv MLP) = 0.776$, $p(CNN > MLP) : 0.224$. Thus at a threshold of $\theta = 0.95$ level we cannot conclude that the two classifiers are practically equivalent or practically different. The standard (frequentist) $t$-test rejects the null hypothesis with a $p$-value of $< 2.2 \times 10^{-16}$ ($t = 9.24$, $df = 9999$). Although the effect size, given by Cohen's $d = 0.0924$ is regarded as "negligible", it is the effect of the $\sqrt{N}$ factor on the $t$-statistic that gives rise to very significant $p$-value for the NHST $t$-test.

However, given the heavy tails indicated by the above analysis, we also used a robustified Student-$t$ model for the $d_j$'s, as discussed above in sec. 6.1. This gave results of $p(CNN < MLP) = 0.00$, $p(CNN \equiv MLP) = 1.00$, and $p(CNN > MLP) = 0.00$ based on 5,000 posterior samples, indicating that the methods should be viewed as equivalent. The difference to the ROPE result obtained with Gaussian assumptions above is

---

[10]The values of $\delta$ were optimized on the test set, which would produce a slight upward bias in the individual scores, but note that they have both been optimized, and we are interested in the *difference* of the two scores.

| count | da-en | de-en | es-en | fr-en | id-en | it-en | nl-en | sv-en | tr-de | tr-en | zh-en |
|-------|-------|-------|-------|-------|-------|-------|-------|-------|-------|-------|-------|
| $n_{00}$ | 18 | 54 | 58 | 79 | 36 | 52 | 68 | 48 | 19 | 39 | 19 |
| $n_{01}$ | 63 | 159 | 175 | 180 | 202 | 185 | 195 | 156 | 64 | 113 | 44 |
| $n_{10}$ | 66 | 198 | 149 | 167 | 167 | 193 | 176 | 133 | 30 | 103 | 46 |
| $n_{11}$ | 183 | 589 | 478 | 574 | 595 | 570 | 561 | 423 | 103 | 298 | 92 |

Table 1: Counts for outcomes $n_{00}$, $n_{01}$, $n_{10}$, and $n_{11}$ for the 11 language pairs. The language labels are Chinese (zh), Danish (da), Dutch (nl), English (en), French (fr), German (de), Indonesian (id), Italian (it), Spanish (es), Swedish (sv) and Turkish (tr).

notable; the robustified method has essentially discounted the tails and focused on the large majority of the differences which are near zero.

### 6.3 Comparing two classifiers on multiple tasks

We illustrate model comparison across multiple tasks (Q3) using data from an experiment on linguistic code-switching. (Code-switching occurs when multilingual speakers use more than one language in an utterance.) Sterner & Teufel (2025a) developed a benchmark containing up to 1,000 minimal pairs of code-switching (CS) sentences for 11 language pairs. Each pair consists of one naturally occurring CS sentence and one minimally manipulated variant, where the variant has the code-switch at an altered location. The classification task for each sentence pair is to identify the naturally occurring CS sentence. In a subsequent paper (Sterner & Teufel, 2025b) the authors compared the performance of a graph neural network (GNN) against a large language model (LLM) for this task.

The results of the experiments are summarized in Table 1. For each language pair there are four counts labelled $n_{00}$, $n_{01}$, $n_{10}$, and $n_{11}$. For example $n_{10}$ indicates the number of test examples where the GNN was correct and the LLM incorrect. As per sec. 5.5 it is only the $n_{01}$ and $n_{10}$ counts that are used in the Bayesian McNemar test (BMcT). A naïve comparison of these numbers shows that the GNN is more accurate than the LLM on 4 of the 11 language pairs.

We first consider carrying out the BMcT for each language pair individually. For example Fig. 5(a) shows the posterior for $\phi$ for language pair 2 (de-en). The posterior mean is around 0.45 indicating that the GNN is outperforming the LLM, but the posterior probabilities are for the three regions are $p(GNN > LLM) : 0.571$, $p(GNN \equiv LLM) : 0.429$, $p(GNN < LLM) : 0.00004$. Hence we cannot conclude that the two classifiers are practically equivalent or practically different for this pair. Fig. 5(b) shows the posterior for language pair 9 (tr-de). In this case the posterior mean is around 0.68, and the posterior region probabilities are $p(GNN > LLM) : 0.000005$, $p(GNN \equiv LLM) : 0.004$, $p(GNN < LLM) : 0.996$. So for this pair we can conclude that the LLM practically outperforms the GNN. In fact tr-de is the only pair where such a conclusion can be made, in all other cases we cannot conclude that the two classifiers are practically equivalent or practically different. Notice that the lower $n_{01}$ and $n_{10}$ counts for the tr-de example relative to the de-en example mean that the posterior variance is greater in Fig. 5(b) than in Fig. 5(a).

For comparison, the standard McNemar's test (`mcnemar.test` in R) gives a $p$-value of 0.045 for language pair 2, and 0.00067 for language pair 9. The $p$-values for all other language pairs are greater than 0.05. So the performance difference on language pair 2 (de-en) would be judged just significant at the $\alpha = 0.05$ level by the standard McNemar's test, but not using the Bayesian McNemar test (with ROPE). The performance difference for language pair 9 (tr-de) is significantly different at the 0.05 level for both the standard and Bayesian McNemar tests.

To estimate the effect size for a given language pair, we first compute $\hat{\phi} = n_{01}/(n_{01} + n_{10})$, and then Cohen's $g = \hat{\phi} - 1/2$. For language pair 9 we have a medium effect of $|g| = 0.18$, and for pair 2 we have $|g| = 0.055$ which just qualifies for the term small. For all other pairs the effect size would be described as negligible.

The above ROPE conclusions are made with the default setting of $\gamma = 0.1$ to define the ROPE. But we can study the effect of varying this parameter, see Fig. 5(c) which plots the probability $p(GNN \equiv LLM)$

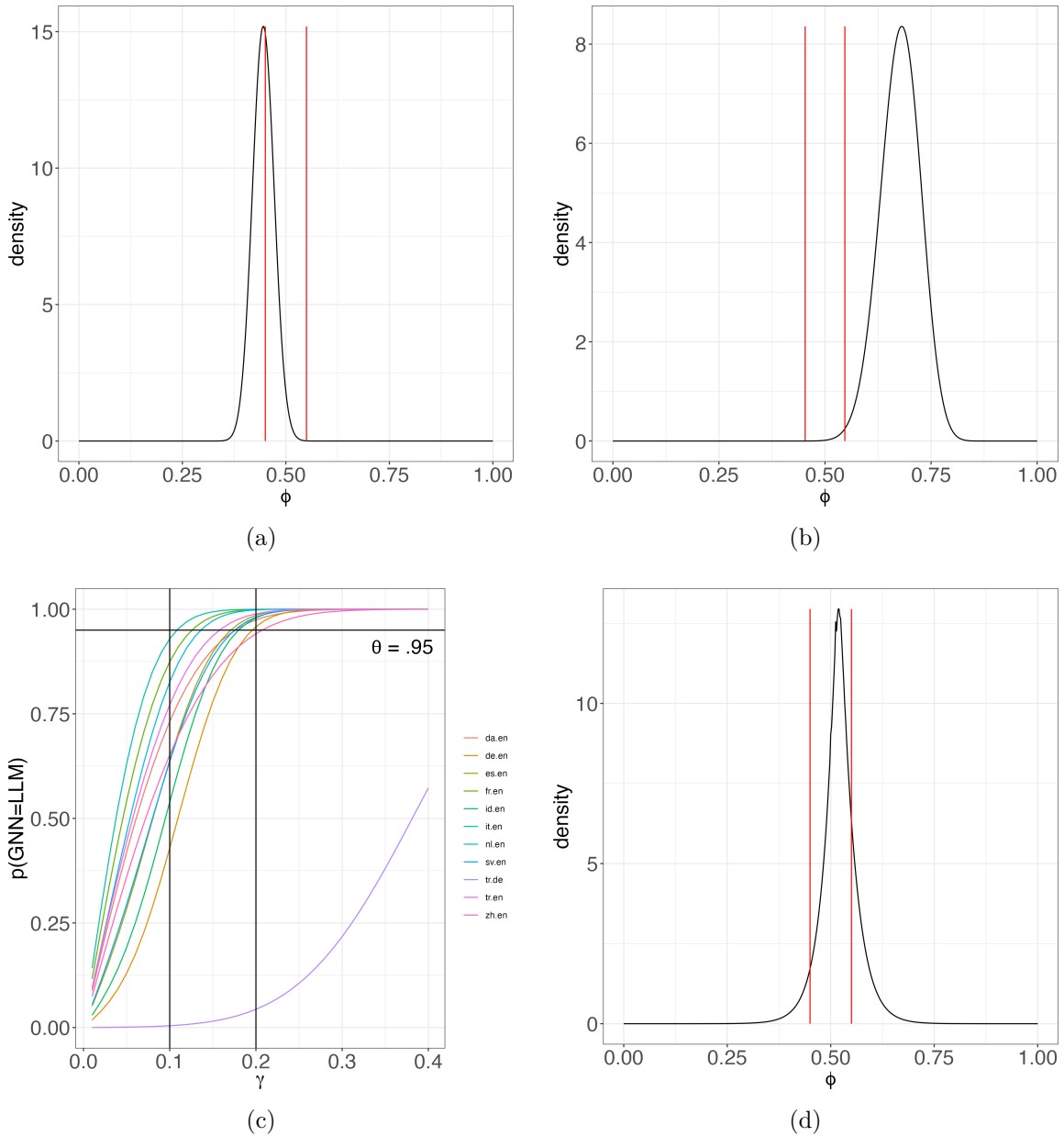

Figure 5: Code switching experiments. (a) Posterior distribution for $\phi$ (black line), and the ROPE boundaries (shown in red) for the de-en language pair. (b) Same for the tr-de language pair. (c) Plot of $p(GNN \equiv LLM)$ against $\gamma$ for all 11 language pairs. (d) The predictive distribution for $\phi_{next}$ (black line), and the ROPE boundaries (shown in red).

against $\gamma$.[11] The horizontal line at 0.95 shows the $\theta$ threshold, while vertical lines at $\gamma = 0.1$ and $\gamma = 0.2$ possible choices for the ROPE width. As expected $p(GNN \equiv LLM)$ increases with $\gamma$, but with $\gamma = 0.1$ for all language pairs except tr-de we cannot conclude that the two classifiers are practically equivalent or practically different. However, for $\gamma = 0.2$ the situation changes dramatically, and of the 10 language pairs excluding tr-de, 9 are now judged practically equivalent, and only zh-en is just below the threshold.

**Scaling the dataset size:** As above, it is interesting to see how the conclusions change as the dataset size is modified. In this case the counts in Table 1 were multiplied by 10. $p$-values from McNemar's test now reject the null hypothesis in 8 out of the 11 cases at the $\alpha = 0.05$ level. As before this illustrates the failure of $p$-values to separate the effect size from the sample size. The three language pairs where $H_0$ was not rejected were da-en, it-en and zh-en; note that in these cases the $n_{01}$ and $n_{10}$ counts are very close, so this makes failure to reject intuitive sense. In these three cases the Bayesian+ROPE method indicates that the GNN and LLM models are practically equivalent at the $\theta = 0.95$ level.

We now turn to the 8 language pairs where the $p$-values were significant. One of these is the tr-de pair where we saw before that the LLM practically outperforms the GNN under the Bayesian+ROPE method; this conclusion still holds (indeed more strongly) with the increased dataset size. For pairs fr-en, nl-en and tr-en, the Bayesian+ROPE method now determines that the two models are practically equivalent at the $\theta = 0.95$ level. For the four remaining language pairs de-en, es-en, id-en and sv-en we cannot conclude that the models are practically equivalent or practically different at the $\theta = 0.95$ level.

**Hierarchical model:** We can also illustrate model comparison across multiple tasks with this dataset, using the hierarchical beta-binomial model discussed in sec. 5.6. This model has hyperparameters $a$ and $b$. One can obtain posterior samples for $a$ and $b$; here we have used the `bang` (Bayesian Analysis, No Gibbs) library in R to do so. Analysis of the samples shows that $a$ and $b$ are very nearly equal, but that there is a large amount of variability in $a + b$; $\log(a + b)$ shows significant variability between values of 3 and 8 around its modal value of 5.28 (data not shown).

The aim of the hierarchical model is to predict $\phi_{next}$, the $\phi$ parameter for the next dataset, given the observations about the 11 language-pairs. The posterior mean $\bar{\phi}_{next}$ (as defined in eq. 9) is equal to 0.521, very close to 0.5. This is not very surprising given the balance between tasks where the GNN wins and cases where the LLM wins, as noted above. Fig. 5(d) shows the predictive distribution, and also the ROPE limits (derived from the standard deviation $s = \sqrt{\bar{\phi}_{next}(1 - \bar{\phi}_{next})}$ and the ROPE $[0.5 - 0.1s, 0.5 + 0.1s]$). The probabilities for the three regions are $p(GNN > LLM) : 0.053$, $p(GNN \equiv LLM) : 0.737$, $p(GNN < LLM) : 0.210$.[12] Thus from the predictive distribution for $\phi_{next}$ based on the 11 observed tasks, we cannot conclude that the two classifiers are expected to be practically equivalent or practically different. For comparison, Friedman's test applied to the 11 pairs of counts gives a $p$-value of 0.366 (Friedman chi-squared $= 0.818$, $df = 1$), and would thus not reject the null hypothesis at the $\alpha = 0.05$ level. But note that this test depends only on the observation that the GNN is more accurate than the LLM on 4 of the 11 language pairs, and does not take into account the closeness of some of the $n_{01}$ and $n_{10}$ counts in Table 1, and possible sampling variation. This aspect is modelled explicitly in the Bayesian hierarchical McNemar test.

# 7 Checklist for best practice

Based on the discussion above, we recommend to:

- Clarify which question (Q1-Q3) you seek to answer.

- Identify a suitable loss function that fits well with the practical application.

---

[11]We thank reviewer dpFV for suggesting this plot and the investigation of the effects of varying $\gamma$.

[12]Note that the analysis in Benavoli et al. (2017, sec. 4.3.2) is slightly different than that given here. For each posterior sample they compute the ROPE probabilities, and then threshold these according to the maximum probability to assign each sample to one of the three regions. These counts are then divided by the sample size to obtain the probabilities shown in their Table 12. In contrast, our results are obtained by averaging the ROPE probabilities over the samples.

- Clearly identify the relevant fixed and random factors of variation that enter into your experiments, and an appropriate experimental design.

- Identify and carry out an appropriate statistical test. We have recommended Bayesian+ROPE tests such as the Bayesian $t$-test, Bayesian McNemar's test, and (for comparisons across multiple tasks) Bayesian hierarchical models. Check the assumptions underlying the selected test, including independence assumptions. Carefully consider appropriate lower and upper limits for the ROPE.

- Report the inferences drawn from the test.

- THINK! It is easy to blindly apply tests, but it is important to maintain a sceptical/critical mindset.

## 8 Discussion

As observed, the current practice on the comparison of machine learning algorithms is often poor. In this paper we have reviewed the issues involved, highlighting the existing body of work on the design and analysis of experiments, including fixed and random effects. There are standard frequentist tests, such as paired $t$-tests, McNemar's test, the Friedman test. However, because of the limitations of NHST as discussed in sec. 5.2, we recommend equivalence testing, and specifically the Bayesian+ROPE approach. In sec. 6 we have provided worked examples of the comparison of machine learning models for both a single task, and over multiple tasks.

What will it take to change current practice? In certain fields, such as medicine and psychology, there are reporting guidelines that must be followed when papers are submitted, see, e.g., the Equator network (Enhancing the QUAlity and Transparency Of health Research)[13] and Appelbaum et al. (2018). We are seeing some moves towards this in the field of machine learning, like the NeurIPS Paper Checklist Guidelines,[14] where item 7 addresses Experiment Statistical Significance. This item mainly covers the reporting of error bars for each algorithm. These are important, but, as highlighted (for example) by Loftus & Masson (1994), they do not take into account the use of paired comparisons. Doing so allows a more powerful comparison of models, which (we argue) is often what is desired. This paper should be seen as contributing towards the creation of such reporting guidelines for work in machine learning and related fields.

As per the final bullet in sec. 7, it is important that authors do not blindly apply tests, but maintain a critical attitude. Thus there should be scope for novel analyses, but in this case the authors will need to justify why standard guidance has not been followed.

### Acknowledgments

We thank the anonymous reviewers whose comments have helped to improve the paper considerably. [Other acknowledgments are redacted to preserve anonymity.]

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

## A    The advantages of paired differences

Consider, for concreteness, the squared error for the prediction $f_A$ of a deterministic predictor $A$ for some test case $(x, y)$. Assume also that the true distribution $p(y|x)$ has mean $\mu_y$ and variance $\sigma_y^2$.[15] We wish to study the squared error $SE_A = (f_A - y)^2$. It is useful to decompose this with respect to the true mean $\mu$, i.e.

$$(f_A - y)^2 = (f_A - \mu_y + \mu_y - y)^2, \tag{11}$$

$$= (f_A - \mu_y)^2 - 2(f_A - \mu_y)(y - \mu_y) + (y - \mu_y)^2. \tag{12}$$

The first *deterministic* term arises from the derivation of $f_A$ from the optimal MSE predictor $\mu_y$. The second and third are stochastic terms depending on $y$. We can compute the expectation of this squared error (or MSE) as

$$MSE_A = \mathbb{E}_y[(f_A - y)^2] = (f_A - \mu)^2 + \mathbb{E}_y[(y - \mu)^2] = (f_A - \mu)^2 + \sigma_y^2. \tag{13}$$

Note that this MSE has an irreducible component due to the variance of $y$. This may vary as a function of $x$, meaning that the scale of the squared error can be different in different locations.

Now consider the paired difference $MSE_A - MSE_B$ between predictors $A$ and $B$ for this same test case. We have that

$$MSE_A - MSE_B = (f_A - \mu)^2 - (f_B - \mu)^2, \tag{14}$$

---

[15]We do not require this distribution to be Gaussian, only to have the given mean and variance.

i.e. the $\sigma_y^2$ term cancels out. Even if we consider the unaveraged difference $SE_A - SE_B$ we have that

$$SE_A - SE_B = (f_A - \mu)^2 - (f_B - \mu)^2 + 2(f_B - f_A)(y - \mu_y). \tag{15}$$

The last term has zero mean, so the difference $SE_A - SE_B$ does not have an additive offset, unlike $SE_A$ (or $SE_B$).

The above analysis was carried out for MSE, but similar (although more complicated) analyses can also be carried out, e.g. for log loss.

