# OpenReview forum: "Use Bayesian Paired Tests with a ROPE to Improve the Comparison of Machine Learning Models"
_TMLR — Under review for TMLR_

### Review · Reviewer_dpFV · 2026-05-04

**Summary Of Contributions:**

The paper advocates replacing frequentist null hypothesis signifance testing NHST with Bayesian paired tests incorporating a region of practical equivalence (ROPE) for comparing and evaluating machine learning models.

The paper list three types of comparisons, each answering different questions:

Q1: comparing Predictors on a single task  (i.e., fixed training set)
Q2: comparing learning algorithms on a single task (i.e., random training set),
Q3: comparing learning algorithms across multiple tasks

The paper presents a Bayesian t-test, a Bayesian McNemar test, and hierarchical extensions for
multi-task settings. Two worked examples are used to illustrate the method, one for comparing a single classifier Q1 on a Mouse-video binary classification task, and another which illustrates Q3, a linguistic code-switching dataset.

The paper concludes with a checklist of recommendations.

The paper clearly positions itself as applying existing techniques rather than a specific technical contribution; however, concretely on my reading these are the contributions:

- The Bayesian t-test + ROPE on log-loss differences (sec. 6.1): this is Benavoli et al. (2017)  Kruschke (2015a) applied to the mouse example. I believe this is identical to prior work only the combination of mouse dataset + this method is new.

- A ROPE construction for the Bayesian McNemar test (sec. 5.4): The definition of $s = sqrt(phi_bar(1 - phi_bar))$ and applying the [0.5 - 0.1s, 0.5 + 0.1s] convention to the Beta posterior from Chechile (2020).

- A hierarchical Beta-Binomial McNemar model (sec. 5.5): A hierarchical model from Gelman et al. (2013, sec. 5.3), applied to the language problem.

Finally, as I undersatnd it, the cornerstone of the 7-point checklist is the specific method advocated in the paper as a replacement of NHST:

"We have recommended Bayesian+ROPE tests such as the
Bayesian t-test, Bayesian McNemar’s test, and (for comparisons across multiple tasks) Bayesian hierarchical models” (sec. 7)


Key strengths:

This is a practically important topic. As the authors point out, model comparison in ML using statistics is something many practitioners struggle with, and addressing this by advancing existing methods using worked-out examples is therefore a worthwhile contribution to TMLR.

The overall structure of the paper is also sound; The distinction Q1-Q3 is helpful and the methods are well explained. The scaling analysis sec 6.1, 6.2 clearly demonstrate how p-values become significant for trivial effect sizes as $N \rightarrow \infty$ while Baysian ROPE converge to practical equivalence.
the Platt (2000) scaling observation where a sigmoid is used to transform output scores illustrates how calibration choices can dominate statistical conclusions. Overall both experiments are well carried out with solid discussions (fixed vs. random factors, anova as a framing)


Key weaknesses:

In my reading, the papers central framing is about applying existing techniques, "accounts of applications of existing techniques" (p 1) rather than methodology. In particular when the application takes the form of a comparison between two schools of thought and advocacy for one of them over the other (Bayesian ROPE compared to NHST, see sec. 5.2, p. 7, "Bayesian analysis" and "Frequentist analysis"; also the checklist; see below), the application must rest on a correct and fair comparison. However, in this case it would be Bayesian ROPE vs. frequentist equivalence testing (TOST; Schuirmann 1987; Lakens 2017) which test the same "is the parameter inside an interval" type of question as ROPE. Since the paper omits this comparison the overall comparison is to my mind not fair and does not present the reader with a good overview. In addition to this, the paper identifies arbitrary choices in NHST as a weakness (p. 8), however, it does not carry out a sensitivity analysis of the ROPE width.

**Additional Comments:**

code:
```
import numpy as np
from scipy import stats
import matplotlib
matplotlib.use("Agg")
import matplotlib.pyplot as plt

data = {
    "de-en": (63, 66),   "da-en": (159, 198), "es-en": (175, 149),
    "fr-en": (180, 167), "it-en": (202, 167), "id-en": (185, 193),
    "nl-en": (195, 176), "sv-en": (156, 133), "tr-en": (64, 30),
    "tr-de": (113, 103), "zh-en": (44, 46),
}
THETA = 0.95
k_fine = np.linspace(0.01, 0.35, 200)

equiv_curves = {pair: [] for pair in data}
n_bayes_equiv_by_k = []
n_tost_equiv_by_k = []

for k in k_fine:
    nb, nt = 0, 0
    for pair, (n01, n10) in data.items():
        a, b = 1 + n01, 1 + n10
        posterior = stats.beta(a, b)
        phi_bar = a / (a + b)
        s = np.sqrt(phi_bar * (1 - phi_bar))
        rope_lo, rope_hi = 0.5 - k * s, 0.5 + k * s
        pe = posterior.cdf(rope_hi) - posterior.cdf(rope_lo)
        equiv_curves[pair].append(pe)
        if pe >= THETA:
            nb += 1
        n = n01 + n10
        p1 = stats.binomtest(n01, n, rope_lo, alternative="greater").pvalue
        p2 = stats.binomtest(n01, n, rope_hi, alternative="less").pvalue
        if p1 < 0.05 and p2 < 0.05:
            nt += 1
    n_bayes_equiv_by_k.append(nb)
    n_tost_equiv_by_k.append(nt)

fig, axes = plt.subplots(1, 2, figsize=(13, 5.5))

ax = axes[0]
cmap = plt.cm.tab10
for i, (pair, pe) in enumerate(equiv_curves.items()):
    ax.plot(k_fine, pe, color=cmap(i % 10), lw=1.5, label=pair)
ax.axhline(THETA, color="black", ls="--", lw=1.2, alpha=0.7)
ax.axvline(0.10, color="red", ls="--", lw=1.5, alpha=0.7)
ax.axvline(0.20, color="orange", ls=":", lw=1.2, alpha=0.7)
ax.text(0.105, 0.02, "paper default\nk=0.10\n(Cohen d=0.2)", color="red", fontsize=7, va="bottom")
ax.text(0.205, 0.02, "k=0.20\n(Cohen d=0.4)", color="orange", fontsize=7, va="bottom")
ax.text(0.35, THETA - 0.03, f"decision threshold = {THETA}", ha="right", fontsize=8, color="black")
ax.set_xlabel("ROPE multiplier k  (ROPE = [0.5 - k*s, 0.5 + k*s])")
ax.set_ylabel("P(equivalent)")
ax.set_title("(a) P(equivalent) vs ROPE width k")
ax.set_xlim(0.01, 0.35)
ax.set_ylim(0, 1.02)
ax.legend(fontsize=6.5, ncol=2, loc="center right")
ax.grid(True, alpha=0.2)

ax = axes[1]
ax.plot(k_fine, n_bayes_equiv_by_k, "-", color="tab:blue", lw=2, label="Bayes+ROPE")
ax.plot(k_fine, n_tost_equiv_by_k, "--", color="tab:orange", lw=2, label="TOST (frequentist)")
ax.axvline(0.10, color="red", ls="--", lw=1.5, alpha=0.7)
ax.axvline(0.20, color="orange", ls=":", lw=1.2, alpha=0.7)
ax.set_xlabel("ROPE multiplier k")
ax.set_ylabel("Pairs declared equivalent (out of 11)")
ax.set_title("(b) Bayes+ROPE vs TOST: same conclusions across all k")
ax.set_xlim(0.01, 0.35)
ax.set_ylim(-0.5, 11.5)
ax.legend(fontsize=9)
ax.grid(True, alpha=0.2)
ax.text(0.105, 11, "paper default\nk=0.10", color="red", fontsize=7, va="top")
ax.text(0.205, 11, "k=0.20", color="orange", fontsize=7, va="top")

fig.suptitle("ROPE width sensitivity: conclusions depend critically on k, and Bayes+ROPE agrees with TOST throughout",
             fontsize=12, y=1.02)
fig.tight_layout()
fig.savefig("rope.png", dpi=150, bbox_inches="tight")
```

**Audience:**

Yes

**Audience Explanation:**

Model comparison is ubiquitous in ML research, and current practice is often poor. The topic is important and the target audience -- ml practitioners who are looking for practical advice -- is large.

However, the specific findings of this paper are largely available in Benavoli et al. (2017) and Kruschke (2015a). The incremental
contributions (ROPE on McNemar, hierarchical McNemar) are modest. As it stands now, two exmaples, in isolation, present a compentent tutorial on Bayesian ROPE.

**Claims And Evidence:**

No

**Claims Explanation:**

I want to begin this section by stating that I have not identified any technical mistakes in the paper, this is a discussion of the "clear evidence" aspect.

I have two main reservations with this manuscript.

The first main reservation is that the paper fundamentally conflates two different lines of argument and thereby presents the reader with at best a confounded comparison and arguably a false dichtomy. This means that the overall recommendation is not convincing, and in fact would be quite misleading to a reader which reads the paper in isolation to understand what to practically do -- i.e., exactly the target audience.

The paper advocate a combination of two things: Bayesian methods over frequentist methods, and testing a parameter falls within an interval of equivalence.

Bayesian and frequentist analysis is presented as two different schools of thought, and the authors present an argument in favor of one over another. This framing is evident throughout the paper:

Abstract: “Due to the limitations of null hypothesis significance testing in
frequentist statistics, Bayesian methods are recommended”

Introduciton, "“these [standard frequenntist methods] are inadequate due to the limitations of null hypothesis significance testing... The solution advanced here is to use Bayesian methods”

main body, “For these reasons we adopt a Bayesian + ROPE estimation approach to the comparison of predictors below” (sec. 5.2).

Discussion (sec. 8): “because of criticisms of NHST as discussed in sec.
5.2, we recommend Bayesian+ROPE approaches”

In particular, I want to point to page. 7, which presents "Bayesian analysis" and "frequentist analysis" as two approaches to the same basic problem (Q1): "Suppose that we are interested in comparing two predictors A and B. ...  Below we describe Bayesian and frequentist analyses for
this situation.". What follows is a "Bayesian analysis", which presents the a test for whether a parameter belongs in an interval of significance (in particular, Bayesian ROPE), folloed by "Frequentist analysis", which presents a null-point NHST for equivalence.


However, *every* criticism of frequentist methods targets point null NHST, i..e H0 : Delta = 0. None of these critisism apply to frequentist equivalence testing (TOST) which targets the same problem as Baysian ROPE: TOST tests whether an effect lies inside an equivalence interval (same as rope), and TOST p-values do not become trivially significant when N increases, rather, they become significantly equivalent which is analogous to ROPE.

Thus, there is a four-way comparision -- the reader has to imagine the table with

** Rows: What to compare: T1: Test if Delta = 0 vs. T2: Delta is within a region of significance

** Columns: How to compare - the statistical approach: Frequentist methods vs. Baeysian methods.

The paper argues Baysian methods + T2 over frequentist methods + T1. However, this comparison is along the diagonal of the table -- this is equivalent to comparing the speed of a motorcycle on a dirt road to a car on a paved road and concluding the car is faster -- it may be true but it is not a well-structured argument. The correct comparison is to argue in favor of testing parameters fall within/outside a region of significance and then present both ROPE or TOST or argue in favor of one over the other.

The authors are aware of equivalence testing, parenthesis on p7, "(The ROPE is somewhat analogous to the
frequentist notion of equivalence testing.)", however, this is the extend of the discussion and TOST is not named, discussed, compared against, and no argument is presented in favor of Bayesian ROPE over TOST.

Furthermore, the paper comes very close to conflating NHST with frequentist, for example when the introduction directly states that frequentist methods “are inadequate due to the
limitations of null hypothesis significance testing.”. I understand that the authors aware of the difference -- for instance sec. 5.1. introduces Cohens d and g which address ASAs principle 5 concern discussed in the paper -- and use these alongside the Bayesian analysis. However, when the main conclusion forcefully argues in favor of Bayesian methods over frequentist methods and frequentist methods are introduced as NHST (and not TOST, or confidence intervals, or any other less naive approach than null-point NHST) one has to read the paper very carefully to avoid the conflation.




The other main reservation is the lack of sensitivity analysis and discussion of k.

For this discussion, I have included python code in the appendix below which will produce a figure with two panels.
These are my attempts at replicating some of the analysis in the paper on the data in table 1. I was unfortunately not able to include the plot the code produces in the review, but here is a link to an anonymous file bucket which will hopefully stay up -- otherwise please just run the code and be aware I may have made errors: https://imgur.com/a/ez5v8Xf

Panel a: In this panel, I have attempted to plot P(equivalent) (y-axis) against the ROPE multiplier k (x-axis) for each of the 11 language pairs, with the 0.95
decision threshold marked as a horizontal dashed line At k=0.1, used in the paper, no pair crosses the 0.95 thresholing at the original data size, so every comparison is inconclusive. as k increaes, pairs progressively cross. as k is scaled from 0.1 to 0.2, 9 of 11 pairs are flipped from inconclusive to equivalent (by analogy with the t-test ROPE convention, this roughly corresponds to treating effects up to Cohen's d = 0.4, i.e. small-to-medium, as negligible). So the conclusion is entirely depending on k.

Panel b: In this panel, I have attempted to plot the number of pair that are equivalent under Bayesian ROPE (y axis, out of 11 languages in total) against k (x-axis) for Bayesian ROPE (solid blue line) and TOST (dashed orange line). The two curves track on top of each other (difference of on pair in some of the region where TOST is slighlty more conservative than Bayesian ROPE) however I believe that this visually shows that the two methods agree in terms of their conclusion and the ROPE width is all that matters in terms of what the conclusion *is*. Note that when producing teh TOST results, I have simply used the ROPE interval [As a side note: if this choice, rather than a fixed interval, significanly affects the TOST conclusion this might present an argumet in favor of ROPE, however, that is outside the scope of this review]


The paper (rightfully) crisisize the arbitrariness of alpha = 0.05: "Scientific conclusions and business or policy decisions should not be based
only on whether a p-value passes a specific threshold" (ASA, quoted in sec. 5.2). The equivalence margin k is arguably more interpretable than alpha since it operates in effect-size units and can in principle be set from domain knowledge (as the authors note in footnote 3). However, the paper does not leverage this advantage -- both examples use the generic k=0.1 convention rather than domain-specific reasoning. Furthermore, k is not specific to the Bayesian framework: TOST requires exactly the same equivalence margin. The choice of k belongs to the equivalence-testing question (row of the 2x2 table), not the Bayesian-vs-frequentist question (column). This makes the lack of sensitivity analysis all the more important, since k drives the conclusions regardless of which framework is used.

The paper adopts k = 0.1 in tewo places:

For the basyesian t-test, section 6.1, "We set the ROPE to [-0.1s, 0.1s] where s denotes the standard deviation of the
difference vector. This follows the advice given in Kruschke (2015b) and Makowski et al. (2025).".

For the Baeysian McNemar test,s ection 5.4,  the standard deviation s = sqrt(phi_bar(1 - phi_bar)) is derived for the Bernoulli
case, but the 0.1 multiplier is not independently justified, but rather carried over from the aformentioned (Krushke 2015b) convention, "one can use the usual ROPE construction of $[0.5 - 0.1s, 0.5 + 0.1s]". The paper does perform post-hoc consistency checks which confirm compatibility with gohesn g trehsolds., but this does not independently justify the 0.1 multipler: "Cohen's $d$ is defined as the difference between two means divided by a standard deviation for the data, $d = m/s$... For calibration Cohen (1988, p. 40) terms a value of $d = 0.2$ as  'small', 0.5 as 'medium' and 0.8 as 'large'.". I think it is worthwhile to point out that Cohen wrote about social sciences and thus what is considered a small effect in social science may not be a small/large effect in ML -- in fact, it would probably depend on the application: Safety critcal ML d=0.2 might be very large, whereas for low stakes applications d=0.5 might not be very large. The point is that k is a parameter of equivalence testing -- shared by ROPE and TOST alike -- and the paper's use of a generic convention rather than domain-specific reasoning means the interpretability advantage of k over alpha is not realized in practice. A sensitivity analysis or domain-specific guidance for choosing k would signifcantly strengthen the presentation.


To summarize, I believe the partial replication raises two concerns:

Firstly, it strenghens the aformentioned point, that TOST is the relevant frequentist benchmark to compare against  (Panel b). It is puzzling why we should reject frequentist methods in favor of Bayesian ROPE when Bayesian ROPE seems to produce the same result as the frequentist method on one of the examples.

Secondly, it highlights that k, for instance k=0.1, is a central calibration parameter. In particular in light of the broad crisism of frequentist calibration parametesr such as alpha, a more in-depth discussion of how to think about, select, or perform a sensitivity analysis of k, similar to alpha in frequentist statistics, would strengthen the presentation.


Other comments:

There are genuinely interesting aspects of Bayesian ROPE. For example, it gives probabilities of all three probability outputs, P(A>B), P(equiv), P(A<B) which could potentially be useful, and it clearly offers more natural extensions to hierarchical models. However, these are subtler merits that require a more naunced argument than what the manuscript currently  makes. Strenghening these points through additional analysis, and highlighting how the methods could be used to revise existing analysis of ML experiments (as the authors note in the introduction, it is not hard to find questionable applications of statistics!) would be valuable.

As a final more personal comment, and not a suggestion for more work, what I have been taught is that it is fine to bold ones table using naive null-point NHST or even "highlighting of the best performing method without any regard for uncertainty" (introduction) as long as one is clear about the choice and the implications, however, ones discussion should be focused around the potential size of an effect and what one has evidence to conclude about it. Bayesian ROPE / TOST is one type of approach which belongs in a toolbox with e.g. frequentist confidence intervals and Baysian credibility intervals -- it is completely valid to focus on ROPE/TOST but it would be worth discussing when it should be used in favor of interval-based analysis rather than a comparision against NHST, which as the the paper correctly points out, produce conclusions which are of very limited practical significance in ML.

**Requested Changes:**

1)  Accurately discuss, and/or compare against, TOST.

I fully concur with the authors conclusion on p. 16, " Thus there should be scope for novel analyses, but in this case the authors will need to justify why standard guidance has not been followed".

To keep the current framing in the list of recommendations, Baeysian ROPE over frequentist methods, the paper must accurately discuss and compare against TOST and demonstrate or argue in favor of bayesian ROPE on those grounds. It is perhaps worthwhile to add to the "Frequentist vs. Bayesian" polemic, however, this topic has been debated so extensively that any additions should present both sides as fairly as possible, which I feel this paper fails to do.

2) Include sensitivity analysis and discuss the practical implications of k.

The current $k=0.1$ is presented with too little justification and no guidance for how to perform a sensitivity analysis. My attempts at replicating the experiments, which may be wrong, suggest changing k from 0.1 to 0.2 flips 9 out of 11 language pairs from inconclusive to equivalent. While k is in principle more interpretable than alpha (it operates in effect-size units), the paper uses a generic convention rather than domain-specific reasoning in both examples, and k is not specific to the Bayesian framework -- TOST requires the same margin. A sensitivity analysis showing how conclusions vary with k is needed.

3) Clarify novelty related to prior work. Although this is more of a tutorial/application, it is much to difficult to understand what is presented as new, and what is an application of prior work, and what is a natural/trivial extension of prior work without a claim of novelty; this distinction is important for the target audience, ML practitioners who are trying to fairly and accurately apply known and well-understood statistical methods to their results.

4) If the paper is presented as a tutorial, expand the example section:

Without a more novel contribution to the Bayesianism vs. Frequentism polemic I don't think the two examples alone represents enough tutorial-material. Expanding the examples section -- either by including real-world examples, bigger examples, more complex test-setups, investigating Q2, or otherwise making it more directly applicable to a ML practitioner with a result table -- is in my view required.


REcommended changes:

1) Add at least one large-scale worked example. Both examples use a limited number of samples. A large-scale example would expand the practical applicability.

2) Discuss sensitiivty to the hypoerparameter choice (a+b)^{-5/2} in McNemar model.

3) Engage more directly with Cross validation. General k-fold cross-validation is tricky and very relevant to the experiment sizes considered here (N in the hundreds). Engaging more directly with this rather than leaving it to future work would srengthen the paper (see impossibility result of Bengio & Grandvalet, 2004, or the cross-validation corrected t-test in Nadeau & Bengio, 2003), in particular when discussing Q2.


Minor:

various smaller mistakes like duplicated words (from from in abstract), LMM -> LLM, standardize reference format.

---

> ### Author Response · Authors · 2026-05-13
> **Responses to reviewer dpFV**
>
> Dear reviewer dpFV
>
> Thank you for your thoughtful and insightful review.
>
> We provide here some brief responses, and a question at the end for the reviewer (and/or action editor).
>
> Full point-by-point responses will be made to accompany the revised manuscript, which we understand should be submitted after all 3 reviews have been received, see
> https://jmlr.org/tmlr/editorial-policies.html .
>
> - We agree that a proper discussion of and comparison with frequentist equivalence testing is needed (e.g. TOST) and this will be added.
>
> - We also agree that a sensitivity analysis of the k parameter in the ROPE $[-k*s, k*s]$ is needed and will be added.
>
> -
> > The Bayesian t-test + ROPE on log-loss differences (sec. 6.1): this is Benavoli et al. (2017) Kruschke (2015a) applied to the mouse example. I believe this is identical to prior work only the combination of mouse dataset + this method is new.
>
> Note that Benavoli et al (2017) actually applied their paired comparisons only on the *aggregate* accuracies on each test fold in k-fold CV, not on individual test examples. So while the examples in
> sec 6. of the submission and in Benavoli et al. (2017) are similar they are not exactly the same; there is no CV in our case. We will clarify this in the revised m/s.
>
> Importantly, our argument (as on p. 1) is that the benchmark setting (development set, fixed test set) is the one that is used almost always in experimental comparisons, and is therefore the one that is
> the most important to address.
>
> - The reviewer states that there is limited technical novelty in the paper. But please note that we are not aware of any work that has applied *paired test-example equivalence testing* to the ML model comparison problem. This framing is developed by a careful review of the questions addressed (sec. 1) and issues of experimental design (sec. 3). This should also be regarded as a contribution to the paper, and is a response to the limitations of the standard point-null NHST paired tests (e.g. paired t-test, McNemar).
>
> - As the reviewer has noted, the comparison of models is a topic of considerable practical importance in ML. Surely there must be a venue in the ML literature for coverage of this topic, and we believe from the stated scope of TMLR that this journal fits the bill.  (Of course issues raised by the reviewers must be addressed to their satisfaction in order to gain acceptance.)
>
> QUESTION: If the reviewer/action editor believes that this paper would be better titled as something like "A Tutorial on ..." we would be open to making that change, and would appreciate your
> input on this point.

---

> ### Comment · Reviewer_dpFV · 2026-05-13
> **Follow-up to the previous response**
>
> Thank you for your thoughtful response. I want to start with the areas of agreement
>
> - Supportive of stronger emphasis on the tutorial aspect; from my perspective the title can stay/change; it can also be emphasized by slightly changing the wording of abstract.
> - I am glad the authors will include a greater discussion of TOST + sensitivity analysis
> - I agree with the clarification on Banavoli 2017 on CV vs. fixed set. The points on Q1/Q2/Q3 framing and benchmark setting is well taken.
>
> My most important concern, which the authors' response does not engage with, is the partial replication of one experiment I included in the review. To quote:
>
> "Panel b: In this panel, I have attempted to plot the number of pair that are equivalent under Bayesian ROPE (y axis, out of 11 languages in total) against k (x-axis) for Bayesian ROPE (solid blue line) and TOST (dashed orange line). The two curves track on top of each other (difference of on pair in some of the region where TOST is slightly more conservative than Bayesian ROPE) however I believe that this visually shows that the two methods agree in terms of their conclusion and the ROPE width is all that matters in terms of what the conclusion is. ".
>
> Also:
>
> "My attempts at replicating the experiments, which may be wrong, suggest changing k from 0.1 to 0.2 flips 9 out of 11 language pairs from inconclusive to equivalent. "
>
> My point is that in this experiment I believe it is the *margin* and not the *statistical framework* which drives the conclusion.
>
> The reason why I think this is significant is that it bears on the papers argument/recomendation of Bayesian methods over frequentist methods, from the abstract ("due to the limitations of null hypothesis significance testing in frequentist statistics, Bayesian methods are recommended") to discussion ("we recommend Bayesian+ROPE approaches") - However, when I apply both Bayesian ROPE and its frequentist counterpart (TOST) to the data in the paper. If the two frameworks agree, and therefore I think the evidence in favor of one over another is weak.
>
> This brings me to another point of the review, the emphasis on *advocacy*. To clarify, I have no objection to a paper that *advocates* Bayesian methods, including in a tutorial that *teaches* Bayesian rope. But a tutorials primary purpose is to inform the reader on best practices, and this places a higher standard on a fair comparison, which fairly describes and weighs relative strengths and weaknesses, than a purely methodological paper. When the paper describes issues with frequentist methods, which includes TOST, and then advocates for Bayesian ROPE, this should be done based on clearly demonstrated advantages, not a comparison against a weaker baseline (point-null NHST). As noted in my review, I do think there are potential strengths to ROPE.
>
> Concretely, in light of the response I (still) recommend:
>
> - Restructure or rethink the Bayesian vs. frequentist comparison so the frequentist side includes equivalence testing, not only null-point NHST. Of course you are welcome to argue against null-point NHST (or Bayesian p-values...); what matters is the comparison of interval-based methods to interval-based methods. See also my discussion of the 2x2 comparison table in the review.
> - Include sensitivity analysis of k that demonstrate how conclusion vary with equivalence margin, and acknowledge this sensitivity in both frameworks. Your discussion of the (arbitrariness) of alpha should be consistent with your discussion of the arbitrariness/objectivity of k.
> - Especially when positioned as tutorial, expand the examples. Two examples are insufficient especially when (If!) one of them does not distinguish the recommended method from its frequentist alternative. Additional examples, especially larger-scale, or engagtement with CV (Q2) would strengthen the tutorial aspect significantly.

---

> > ### Author Response · Authors · 2026-05-29
> > **Response to Follow-up to the previous response -- part 1**
> >
> > We once again thank the reviewer for their thoughtful and insightful comments.
> >
> > We have delayed responding in order to learn more about frequentist and Bayesian equivalence testing, and their comparison. We had seen mention of frequentist equivalence testing before in Kruschke (2015), but had not explored it in any depth.  For frequentist equivalence testing the book by Wellek (2010) is a very useful reference. With respect to the comparison of frequentist and Bayesian approaches we have found the paper by Linde et al. (2023) "Decisions About Equivalence: A Comparison of TOST, HDI-ROPE, and the Bayes Factor" [LTSWvR23] to be helpful, along with criticism by Campbell and Gustafson (2024) [CG24] and a reply by Linde et al. (2024) [LTWvR24].
> >
> > > My most important concern, which the authors' response does not engage with, is the partial replication of one experiment I included in the review. To quote:
> >
> > > "Panel b: In this panel, I have attempted to plot the number of pair that are equivalent under Bayesian ROPE (y axis, out of 11 languages in total) against k (x-axis) for Bayesian ROPE (solid blue line) and TOST (dashed orange line). The two curves track on top of each other (difference of on pair in some of the region where TOST is slightly more conservative than Bayesian ROPE) however I believe that this visually shows that the two methods agree in terms of their conclusion and the ROPE width is all that matters in terms of what the conclusion is. ".
> >
> > Note that the Bayes factor method in [LTSWvR23] is similar to the ROPE-(full) procedure (assessing the posterior mass in the ROPE), except for the priors for the ROPE hypothesis and its complement.
> >
> > We will expand the coverage of equivalence testing in the paper to cover specification of the ROPE, Bayesian analysis, frequentist analysis, and a comparison of Bayesian and frequentist equivalence testing. BTW [CG24] note that TOST is not in fact the optimal frequentist test, and they also use the optimal test (OT) as proposed by Romano (2005). They comment that TOST has little power to establish equivalence when the sample size or margin are small.
> >
> > We now give an argument why the TOST, HDI-ROPE and ROPE-(full) procedures can give similar results.
> >
> > [LTSWvR23] consider the HDI-ROPE procedure, as recommended in Kruschke's book (2015) sec 12.1.  The highest density interval (HDI) for the posterior is the span of values that are most credible and contain $1−\alpha$ of the mass of the distribution. This credible interval is a Bayesian analogue to a frequentist confidence interval (CI). The decision process in then to declare equivalence if the ROPE completely contains the $1-\alpha$ HDI of the posterior.
> >
> > One can also think of the TOST procedure in terms of a confidence interval (see, e.g., Schuirmann 1987). The null hypothesis for equivalence testing (i.e. that $\Delta$ lies to the left or right of the ROPE) is rejected at significance level $\alpha$ if the $1− 2 \alpha$ confidence interval fully lies within the ROPE.
> >
> > Despite the differences in frequentist and Bayesian philosophy, the HDI credible interval and the frequentist confidence interval (CI) can sometimes be similar in practice.  For example [Jaynes76] showed that for a single unknown location parameter having a uniform prior, the (Bayesian) credible interval and the CI coincide. Although the TOST confidence interval is for $1− 2 \alpha$ and the HDI-ROPE credible interval is for $1 - \alpha$, these two intervals may be quite similar in
> > practice. Hence we would expect the HDI credible interval and the CI to be similar for the code switch experiments, as the beta(1,1) prior is non-informative, and the counts involved are considerably larger than 1.  Note also that the HDI contains 95% of the probability mass
> > of the posterior, and if it lies entirely within the ROPE (the decision criterion for declaring equivalence under the HDI-ROPE procedure), then at least 95% of the posterior mass must lie within the ROPE, and thus be judged equivalent under the ROPE-(full) procedure for $\theta = 0.95$. Hence it is not surprising (at least in retrospect) that the TOST and HDI-ROPE and ROPE-(full) can give similar results. But note that [LTSWvR23] recommend the Bayes factor approach, particularly for when the sample size is relatively small.
> >
> > > I believe that this visually shows that the two methods agree in terms
> > of their conclusion and the ROPE width is all that matters in terms of
> > what the conclusion is. ".
> >
> > As stated previously we agree that a sensitivity analysis of the $k$ parameter in the ROPE $[-k*s, k*s] $ is needed and will be added.

---

> > > ### Author Response · Authors · 2026-05-29
> > > **Response to Follow-up to the previous response -- part 2**
> > >
> > > > My point is that in this experiment I believe it is the margin and not the statistical framework which drives the conclusion.
> > >
> > > > The reason why I think this is significant is that it bears on the papers argument/recommendation of Bayesian methods over frequentist methods, from the abstract ("due to the limitations of null hypothesis significance testing in frequentist statistics, Bayesian methods are recommended") to discussion ("we recommend Bayesian+ROPE approaches")
> > > > However, when I apply both Bayesian ROPE and its frequentist counterpart (TOST) to the data in the paper. If the two frameworks agree, and therefore I think the evidence in favor of one over another is weak.
> > >
> > > We agree that the abstract and other places in the text need to be revised. As pointed out by the reviewer in their first set of comments, one can think of a 2x2 table with two axes, (1) Delta = 0
> > > vs. Delta is within the ROPE and (2) frequentist vs Bayesian methods.
> > >
> > > We will add a discussion of how Bayesian analysis of the point null hypothesis is done (Bayes factors etc). However, there has been much discussion about the use of the point null hypothesis in Bayesian hypothesis testing.  For example Gelman et al. (2013, p. 95) state “In problems involving a continuous parameter $\theta$ (say the difference between two means), the hypothesis that $\theta$ is exactly zero is rarely reasonable”. Benavoli et al. (2017, sec. 3) make the same point, and thus follow the parameter estimation plus equivalence testing approach
> > > rather than Bayesian model comparison. We agree with this viewpoint.
> > >
> > > The discussion of [LTSWvR23] does argue for the ROPE-(full) method, although we do agree that in some cases the results of the TOST (and/or OT) methods and ROPE-(full) will be similar, and will mention this. Other reasons for preferring the Bayesian approach are as pointed
> > > out by the reviewer in their first review: the direct access to the $p(A<B)$, $p(A \equiv B)$, and $p(A>B)$ probabilities it affords, and hierarchical modelling for the multitask case.
> > >
> > > > This brings me to another point of the review, the emphasis on advocacy. To clarify, I have no objection to a paper that advocates Bayesian methods, including in a tutorial that teaches Bayesian rope. But a tutorials primary purpose is to inform the reader on best
> > > practices, and this places a higher standard on a fair comparison, which fairly describes and weighs relative strengths and weaknesses, than a purely methodological paper. When the paper describes issues with frequentist methods, which includes TOST, and then advocates for
> > > Bayesian ROPE, this should be done based on clearly demonstrated advantages, not a comparison against a weaker baseline (point-null NHST). As noted in my review, I do think there are potential strengths to ROPE.
> > >
> > > We agree with these points, see the discussion above.
> > >
> > > > Restructure or rethink the Bayesian vs. frequentist comparison so the frequentist side includes equivalence testing, not only   null-point NHST. Of course you are welcome to argue against
> > >  null-point NHST (or Bayesian p-values...); what matters is the  comparison of interval-based methods to interval-based methods. See   also my discussion of the 2x2 comparison table in the review.
> > >
> > > Yes this will be done, as discussed above.
> > >
> > > > Include sensitivity analysis of k that demonstrate how conclusion vary with equivalence margin, and acknowledge this sensitivity in   both frameworks. Your discussion of the (arbitrariness) of alpha   should be consistent with your discussion of the
> > > arbitrariness/objectivity of k.
> > >
> > > A sensitivity analysis wrt $k$ will be added.
> > >
> > > > Especially when positioned as tutorial, expand the examples. Two  examples are insufficient especially when (If!) one of them does not   distinguish the recommended method from its frequentist   alternative. Additional examples, especially larger-scale, or engagement with CV (Q2) would strengthen the tutorial aspect  significantly.
> > >
> > > We will expand the examples to include larger data scale, using the classic MNIST dataset (60k training examples, 10k test). But note that the examples were chosen to illustrate the comparison of a real-valued quantity, the log probability of the correct label (Ex 1), and 0/1 loss
> > > (Ex 2). Ex 2 also addresses the multitask case. These two examples thus already provide illustrations of the Bayesian t-test (sec 5.3), Bayesian McNemar test (sec 5.4) and Bayesian hierarchical models (sec 5.5).
> > >
> > > [references will appear in the next response due to 5000 chars limit]

---

> > > > ### Author Response · Authors · 2026-05-29
> > > > **Response to Follow-up to the previous response --- references**
> > > >
> > > > [CG24] H. Campbell and P. Gustafson. The Bayes Factor, HDI-ROPE and Frequentist Equivalence Tests Can All Be Reverse Engineered–Almost Exactly–From One Another: Reply to Linde et al. (2021). Psychological Methods, 29(3):613–623, 2024.
> > > >
> > > > [Jaynes76]
> > > > E. T. Jaynes. Confidence Intervals vs Bayesian Intervals. In W. L. Harper and C. A. Hooker (eds.), Foundations of Probability Theory, Statistical Inference, and Statistical Theories of Science,
> > > > volume II, pp. 175–257. D. Reidel, 1976.
> > > >
> > > > [LTSWvR23]
> > > > M. Linde, J. N. Tendeiro, R. Selker, E.-J. Wagenmakers, and D. van Ravenzwaaij. Decisions About Equivalence: A Comparison of TOST, HDI-ROPE, and the Bayes Factor. Psychological Methods, 28(3):740–755, 2023.
> > > >
> > > > [LTWvR24]
> > > > M. Linde, J. N. Tendeiro, E.-J. Wagenmakers, and D. van Ravenzwaaij. Practical Implications of Equating Equivalence Tests: Reply to Campbell and Gustafson (2022). Psychological Methods,
> > > > 29(3):603–605, 2024.
> > > >
> > > > [Romano05]
> > > > J. P. Romano. Optimal Testing of Equivalence Hypotheses. The Annals of Statistics, 33:1036-1047, 2005.
> > > >
> > > > D. J. Schuirmann. A Comparison of the Two One-Sided Tests Procedure and the Power Approach for Assessing the Equivalence of Average Bioavailability. Journal of Pharmacokinetics and Biopharmaceutics, 15:657–680, 1987.
> > > >
> > > > [Wellek10]
> > > > S. Wellek. Testing Statistical Hypotheses of Equivalence and Noninferiority. CRC Press, second edition, 2010.

---

> ### Author Response · Authors · 2026-06-15
> **Responses to reviewer dpFV wrt the revised m/s**
>
> Again we thank the reviewer for their helpful and insightful comments.
>
> > I have two main reservations with this manuscript.
>
> > The first main reservation is that the paper fundamentally conflates
> two different lines of argument and thereby presents the reader with
> at best a confounded comparison and arguably a false dichotomy. This
> means that the overall recommendation is not convincing, and in fact
> would be quite misleading to a reader which reads the paper in
> isolation to understand what to practically do -- i.e., exactly the
> target audience.
>
> > Bayesian and frequentist analysis is presented as two different
> schools of thought, and the authors present an argument in favor of
> one over another. This framing is evident throughout the paper:
>
> > However, every criticism of frequentist methods targets point null
> NHST, i..e H0 : Delta = 0. None of these criticism apply to
> frequentist equivalence testing (TOST) which targets the same problem
> as Bayesian ROPE: TOST tests whether an effect lies inside an
> equivalence interval (same as rope), and TOST p-values do not become
> trivially significant when N increases, rather, they become
> significantly equivalent which is analogous to ROPE.
>
> > Requested Changes: 1. Accurately discuss, and/or compare against, TOST
>
> In line with the reviewer's criticisms, the paper has been revised to
> compare and contrast *point null* testing (sec 5.2) against
> equivalence testing (sec 5.3). Sec 5.2 now includes (and argues
> against) Bayesian null hypothesis testing.
>
> Sec 5.3 discusses equivalence testing, including specifying the ROPE,
> Bayesian analysis, frequentist analysis (including TOST), and
> comparison of Bayesian and frequentist equivalence testing. As per our
> previous responses, we argue for Bayesian over frequentist equivalence
> testing, based on Linde et al. (2023), Campbell & Gustafson (2024),
> and Linde et al. (2024), but do acknowledge that the results of the
> different methods can be similar.
>
> We have also revised the introductory text on p 1 to reflect these changes.
>
> > The other main reservation is the lack of sensitivity analysis and
> discussion of k.
>
> > Requested changes: 2. Include sensitivity analysis and discuss the practical implications of k.
>
> Discussion of varying the width of the ROPE by a factor $\gamma$
> (equivalent to the reviewer's $k$) has been added in the section of
> 5.3 on Specifying the ROPE. We have added examples of this in Fig 3(c)
> for the MNIST example, and for the code switching example in Fig
> 5(c). We credit the reviewer for suggesting this latter plot and the
> investigation of the effects of varying $\gamma$ in footnote 11.
>
> > Requested changes: 3. Clarify novelty related to prior work.
>
> The text on pages 1 and 2 has been expanded to highlight differences
> of this paper to Benavoli et al. (2017), in that BCDZ used a
> cross-validation approach, but used comparisons of the *average*
> performance on each cross-validation test fold, not on each individual
> test example.
>
>
> > Requested changes: 4. If the paper is presented as a tutorial,
>   expand the example section:
>
> > Add at least one large-scale worked example. Both examples use a
> limited number of samples. A large-scale example would expand the
> practical applicability.
>
> We have added the new example in sec 6.2 on the 10 class MNIST
> digit classification problem. This has a lot more examples (60,000
> training, 10,000 test). We have also used ensembling of the CNN and MLP
> predictors to illustrate handling of weight initialization variability.

---

### Review · Reviewer_jnT8 · 2026-05-27

**Summary Of Contributions:**

## Strengths

- As statistical hypothesis testing, Bayesian methods are preferred to frequentist methods (commonly known null hypothesis significance testing) in several aspects, for example, it does not tend to judge that the difference exists only because the number of samples are increased. Introducing it to comparisons of machine learning models is useful.
- ROPE (region of practical equivalence), the concept already known in Bayesian testing, is calculated for comparisons of machine learning models.

## Weaknesses

- The effect of the choice of the prior (since it is Bayesian methods) is desired to be examined.
- Experimental results looks limited.

**Audience:**

Yes

**Audience Explanation:**

Since ordinary null hypothesis significance testing using frequentist probability encounters several common problems, the solution using Bayesian probability will help many statistical inferences.

**Broader Impact Concerns:**

No particular concern.

**Claims And Evidence:**

Yes

**Claims Explanation:**

Although the reviewer desire more things to be examined (shown in "Summary Of Contributions" -> "Weaknesses"), the claims written in the paper seems to be shown.

**Requested Changes:**

## Points critical to securing my recommendation for acceptance

- Section 5.2: (Although it may not be a problem of this paper but the problem of Bayesian statistical hypothesis testing itself,) as far as my understanding, the ROPE region is a hyperparameter determined by prior knowledge. However, I felt that such existence of hyperparameters makes this statistical hypothesis testing difficult to assess the validity since it significantly depends on the region. Are there any consensus about how to set ROPE region (or any priors needed for these tests) for practical uses? Or are there any way to determine the ROPE region depending only on the data?
- Section 5.3: Is the saying "We assume that $d\_j \\sim N(\\Delta, \\sigma^2)$" means that the distribution of the loss is assumed to be produce this distribution? If so, what losses satisfy this?
- Section 5.5: What does the sentence "The aim of the hierarchical model is to predict $\\phi\_{next}$, the $\\phi$ parameter for the next dataset drawn from the same underlying distribution as the q datasets" intend? Does $\\phi\_{next}$ means "one of $\\phi\_1$, $\\phi\_2$, ..., $\\phi\_q$ computed next"? If so, does it intend that we compute the conditional distribution of $\\phi\_{next}$ conditioned by the known ones?
- Section 5.5: Although the concept of the term "context" is defined in Section 1, but how it is defined mathematically is unclear, even if reading (Lacoste et al., 2012). Does the "context" means that, since each task is represented as respective data distributions, the "context" stores multiple such distributions (i.e., multiple tasks)? Ideally, it should be defined purely mathematically.
- Section 6 overall: For each experiment only one dataset is used, but why? Especially, to show the theoretical properties of the method, please consider using multiple datasets.
- Section 6 overall: In order to evaluate practical availability of the method (or any Bayesian statistical hypothesis testing), sensitivity analysis for the choice of the prior is desired.

## Points that simply strengthen the work in my view

- Section 3: Although it may be related to the experimental design problem, I felt introducing the experimental design problem becomes confusing, since the discussion of what components are random for Q1 to Q3 can be discussed regardless of the experimental design problem.
- Section 5.6: It states that "I.e., by better modelling of the situation with a posterior on $\\delta$, and the use of the ROPE, the issue of multiple comparisons is mitigated", but is the algorithm of Section 5.5 built on this concept? If so, please add brief explanation about this.
- Section 6.1: What are $\\log p(LgR)$, $\\log p(SVM)$ or the like? Please write mathematical definitions.
- Sections 6.1 and 6.2, subsection "Scaling the dataset size": Does it intend that, when the sample size is increased, (frequentist) p-value tends to be decreased (i.e., it is likely to be judged that the difference exists), while ROPE tends to be increased (i.e., conversely, it is likely to be judged that the difference does not exist)?

---

> ### Author Response · Authors · 2026-06-02
> **Response to comments from reviewer jnT8 -- part 1**
>
> We thank the reviewer for their helpful comments.
>
> We provide here some brief responses.  Full point-by-point responses
> will be made to accompany the revised manuscript, which we understand
> should be submitted after all 3 reviews have been received, see
> https://jmlr.org/tmlr/editorial-policies.html .
>
> **Weaknesses**
>
> > * The effect of the choice of the prior (since it is Bayesian methods)
> is desired to be examined.
>
> See below.
>
> > * Experimental results looks limited.
>
> See below.
>
> **Requested Changes:**
>
> > * Section 5.2: (Although it may not be a problem of this paper but the
>   problem of Bayesian statistical hypothesis testing itself,) as far
>   as my understanding, the ROPE region is a hyperparameter determined
>   by prior knowledge. However, I felt that such existence of
>   hyperparameters makes this statistical hypothesis testing difficult
>   to assess the validity since it significantly depends on the
>   region. Are there any consensus about how to set ROPE region (or any
>   priors needed for these tests) for practical uses? Or are there any
>   way to determine the ROPE region depending only on the data?
>
> A new section on equivalence testing will be added, including more discussion of setting the ROPE. If the test quantity is interpretable then ideally we should be able to set the ROPE based on our knowledge of the problem. For example the US Food and Drug Administration (FDA)
> guidance for drug bioequivalence is for an 80-125% range of efficacy of a new drug relative to reference drug.
>
> As mentioned on p 7. of the m/s,  (Kruschke, 2015b; Makowski et al., 2025) recommend the choice of $\pm 0.1s$ which is what we have used. However, it is certainly worthwhile to investigate the sensitivity of the conclusions to this choice, and this will be done in the revised m/s.
>
> > * Section 5.3: Is the saying "We assume that d_j ~ N(Delta, sigma^2)"
> means that the distribution of the loss is assumed to be produce this
> distribution?  If so, what losses satisfy this?
>
> The $d_j$s depend on the predictors and datasets etc. We know of no results on showing that the losses are Gaussian for some situation. The Gaussian assumption is the obvious choice from the ANOVA model. However, as mentioned in sec 6.1, we can investigate this assumption,
> e.g. by making a Q-Q plot of the $d_j$s, and if they are found to not conform well to a Gaussian distribution, we can use alternatives such as a robustified model with a Student-t distribution instead of the Gaussian.
>
>
> > Section 5.5: What does the sentence "The aim of the hierarchical
>   model is to predict $\phi_{next}$, the parameter for the next dataset drawn from
>   the same underlying distribution as the q datasets" intend? Does
>   means "one of phi_1, phi_2, ..., phi_q computed next"? If so, does
>   it intend that we compute the conditional distribution of phi_next
>   conditioned by the known ones?
>
> It is intended to mean "the  conditional distribution of $\phi_{next}$ given $\mathbf{d_1}, \mathbf{d_2}, \ldots, \mathbf{d_q}$. We will clarify this the text. Also if $\phi_{q+1}$ would be clearer than $\phi_{next}$ we are happy to make that change.
>
> > Section 5.5: Although the concept of the term "context" is defined
>   in Section 1, but how it is defined mathematically is unclear, even
>   if reading (Lacoste et al., 2012). Does the "context" means that,
>   since each task is represented as respective data distributions, the
>   "context" stores multiple such distributions (i.e., multiple tasks)?
>   Ideally, it should be defined purely mathematically.
>
> Lacoste et al. (2012) define a context by saying "It represents a distribution over the different tasks a learning algorithm is meant to encounter." As far as we are aware, a more mathematical definition would be via a hierarchical model, as discussed e.g., in chapter 5 Gelman et al. (2013) (cited in our sec. 5.5).
>
> > Section 6 overall: For each experiment only one dataset is used, but
>   why? Especially, to show the theoretical properties of the method,
>   please consider using multiple datasets.
>
> We will add a further example from the MNIST digit classification problem. But note that the main aim is to illustrate the use of the Bayesian t-test, the Bayesian McNemar test and Bayesian hierarchical tests. The revised paper will be around 25 pages long and there are
> diminishing returns for adding more examples.
>
> > Section 6 overall: In order to evaluate practical availability of
>   the method (or any Bayesian statistical hypothesis testing),
>   sensitivity analysis for the choice of the prior is desired.
>
> Yes this is true, but the priors used here are in general non-informative, and are not intended to be subjective unless there is prior knowledge that can be used.  For example in the derivation of eq 4, non-informative priors on $\Delta$ and $\sigma^2$ are used, and the flat beta(1,1) prior is used as a prior on $\phi$ in sec 5.4.
>
> We will add a comment on the issue of priors in the m/s.
>
> [responses continue in part 2, due to 5000 chars limit]

---

> > ### Author Response · Authors · 2026-06-02
> > **Response to comments from reviewer jnT8 -- part 2**
> >
> > **Points that simply strengthen the work in my view**
> >
> >   > Section 3: Although it may be related to the experimental design
> >   problem, I felt introducing the experimental design problem becomes
> >   confusing, since the discussion of what components are random for Q1
> >   to Q3 can be discussed regardless of the experimental design
> >   problem.
> >
> > We beg to differ here. We found that the discipline of considering factors of variation and an ANOVA-type model to clarify our thinking. Also, for example, it allows us to identify the "repeated measures" situation (aka within-subjects design in human
> > subjects). The problem of comparing methods is universal in science, and we feel the framing in sec. 3 helps to show this.
> >
> >
> > > Section 5.6: It states that "I.e., by better modelling of the
> >   situation with a posterior on $\Delta$, and the use of the ROPE, the issue
> >   of multiple comparisons is mitigated", but is the algorithm of
> >   Section 5.5 built on this concept? If so, please add brief
> >   explanation about this.
> >
> > Gelman et al. (2012) identify two key diﬀerences from the classical perspective to the approach described in their paper. First, they say "we are typically not terribly concerned with Type 1 error because we rarely believe that it is possible for the null hypothesis to be strictly true", as quoted in the m/s. Secondly they say, "we believe
> > that the problem is not multiple testing but rather insuﬃcient modeling of the relationship between the corresponding parameters of the model", and point to Bayesian hierarchical (or multilevel) modelling to address this.
> >
> > We have already argued for the rejection of point null hypotheses in sec 5.2, addressing their first point. This leads to the better modelling of the situation with a posterior on $\Delta$, and the use of the ROPE to avoid a point null hypothesis.
> >
> > In sec 5.7 we also mention that "It would be possible to consider a hierarchical model over algorithms (as opposed to tasks, as in sec. 5.6)", but leave that to future work.  The key point is that the most straightforward question is to ask about the comparison of two methods, and this is what practitioners usually want to do (in our experience).
> >
> >
> > > Section 6.1: What are log p(LgR),  log p(SVM) or the like? Please
> >    write mathematical definitions.
> >
> > Apologies, this should be $\log p^{LgR}_j$, the log probability of test case $j$ under the LgR model, and similarly for the SVM. This will be fixed in the revised m/s.
> >
> > > Sections 6.1 and 6.2, subsection "Scaling the dataset size": Does it
> > intend that, when the sample size is increased, (frequentist) p-value
> > tends to be decreased (i.e., it is likely to be judged that the
> > difference exists), while ROPE tends to be increased (i.e.,
> > conversely, it is likely to be judged that the difference does not
> > exist)?
> >
> > The ROPE does not depend on the sample size $N$. For example in the specification as $\pm 0.1s$, the $s$ is the standard deviation of the $d_j$s, not the standard error of the mean. However, the posterior will typically tighten as the sample size is increased. Depending on where the posterior mode is located (assuming it is unimodal), we would expect the posterior mass to concentrate one of the three
> > regions $(A > B)$, $(A \equiv B)$, $(A < B)$ as $N$ gets large.

---

> > > ### Comment · Reviewer_jnT8 · 2026-06-04
> > >
> > > Thank you for rigorous responses. Let me present additional comments on the responses.
> > >
> > > > Lacoste et al. (2012) define a context by saying "It represents a distribution over the different tasks a learning algorithm is meant to encounter." As far as we are aware, a more mathematical definition would be via a hierarchical model, as discussed e.g., in chapter 5 Gelman et al. (2013) (cited in our sec. 5.5).
> > >
> > > I understood the mathematical definition of hierarchical models, but not the term "context" itself. Is it "a distribution", not a family of distributions? If it is single distribution, how is it defined for different tasks?
> > >
> > > > We will add a further example from the MNIST digit classification problem. But note that the main aim is to illustrate the use of the Bayesian t-test, the Bayesian McNemar test and Bayesian hierarchical tests.
> > > > Yes this is true, but the priors used here are in general non-informative, and are not intended to be subjective unless there is prior knowledge that can be used. For example in the derivation of eq 4, non-informative priors on $\\Delta$ and $\\sigma^2$ are used, and the flat beta(1,1) prior is used as a prior on in sec 5.4.
> > >
> > > If the main interest is to demonstrate the use of these Bayesian tests, then I think that presenting just results is insufficient; presenting how to replace (frequentist) tests is desired, including choices of priors (or other hyperparameters). Otherwise it is unclear how the methods are available and effective.

---

> ### Author Response · Authors · 2026-06-13
> **Response to Official Comment by Reviewer jnT8**
>
> > I understood the mathematical definition of hierarchical models, but
> not the term "context" itself. Is it "a distribution", not a family of
> distributions? If it is single distribution, how is it defined for
> different tasks?
>
> The concept of a number of related ML tasks comes up a lot, e.g. in the multi-task learning (MTL) literature. See e.g., the review paper "A Survey on Multi-Task Learning", Yu Zhang and Qiang Yang, IEEE Transactions on Knowledge and Data Engineering, vol. 34, no. 12, pp. 5586-5609, 1 Dec. 2022, doi: 10.1109/TKDE.2021.3070203. This paper discusses many different MTL models for multi-task learning, but does not seem to offer a mathematical definition of
> what related tasks are. We presume the reviewer is looking for some connection/similarity between, say, $p_{T_1}(x_1,y_1)$ and $p_{T_2}(x_2,y_2)$ on tasks $T_1$ and $T_2$. We are not aware of such definitions, but it is clear that the concept of a number of related tasks (aka a "context") is widely used in the ML literature, and that is all we need for the current paper.
>
>
> > If the main interest is to demonstrate the use of these Bayesian
> tests, then I think that presenting just results is insufficient;
> presenting how to replace (frequentist) tests is desired, including
> choices of priors (or other hyperparameters). Otherwise it is unclear
> how the methods are available and effective.
>
> The worked examples in sec. 6 are intended to show how frequentist tests may be replaced by Bayesian+ROPE tests, using standard choices for priors and other hyperparameters

---

### Review · Reviewer_ngdw · 2026-06-02

**Summary Of Contributions:**

The paper argues that ML model comparison is often statistically under-specified, and advocates paired Bayesian tests with a region of practical equivalence (ROPE) as a more informative alternative to unpaired error bars and NHST-based significance testing. It organizes comparison settings using Dietterich's taxonomy, reviews relevant frequentist tests, presents Bayesian paired $t$-tests, Bayesian McNemar tests, and hierarchical Bayesian extensions for multiple tasks, and illustrates these methods through two worked examples and a reporting checklist.

The main strengths are the paper's practical motivation, pedagogical exposition, emphasis on paired comparisons and fixed/random factors, and examples. The main weaknesses are limited methodological novelty relative to prior Bayesian and ROPE work, under-discussed choices of ROPE thresholds and priors, and relatively limited empirical validation or sensitivity analysis of the proposed Bayesian McNemar/ hierarchical extensions.

**Audience:**

Yes

**Audience Explanation:**

The paper should be of interest to at least some of TMLR’s audience, especially researchers who run, interpret, or review empirical ML benchmarks. Its contribution is a practical synthesis and modest extension of Bayesian paired testing with ROPE for common model-comparison settings. The discussion of paired designs, fixed and random factors, Bayesian t-tests, Bayesian McNemar tests, hierarchical multi-task comparisons, worked examples, and the reporting checklist are relevant to improving empirical evaluation practice in ML.

The likely audience is smaller than for a major new modeling method, and the methodological novelty is limited. However, statistically sound model comparison is central enough to empirical ML that some TMLR readers would benefit from the paper's findings and recommendations.

**Broader Impact Concerns:**

I do not see broader impact concerns that would require a separate broader impact statement. The paper is primarily methodological guidance on statistical model comparison and reporting practice. Its likely impact is to improve reliability and transparency in empirical ML evaluation. I do not see direct ethical risks beyond the usual need to avoid overclaiming from statistical tests.

**Claims And Evidence:**

Yes

**Claims Explanation:**

The paper’s claims are supported. The core methodological claims are backed by statistical literature, derivations for why paired comparisons reduce irrelevant variance, and examples illustrating Bayesian plus ROPE analyses for single-task and multi-task model comparison. The examples are clear and support the paper’s message, especially the distinction between statistical significance and practical equivalence as sample size changes.

The evidential support is stronger for the tutorial/statistical claims than for the broader claims about current ML practice. The claim that model comparison is often poorly handled is plausible, but is motivated mostly anecdotally. Some proposed defaults, such as the ROPE choice and the Bayesian McNemar/hierarchical extensions, would also benefit from more sensitivity analysis or calibration. Overall, I would judge the claims as clearly argued, with some limitations in how comprehensively they are empirically supported.

**Requested Changes:**

I do not see any required changes that would be critical to securing my recommendation.

1. Clarify the incremental contribution relative to Benavoli et al. (2017), especially what is new beyond synthesis/tutorial exposition.
2. Add more discussion or sensitivity analysis for the ROPE choice, prior choices, and the proposed Bayesian McNemar/hierarchical variants.
3. Either substantiate the claim that current ML practice is poor, or soften it as anecdotal motivation.
4. Give more practical guidance on common violations of independence assumptions, including correlated examples and seed/initialization variability.

---

> ### Author Response · Authors · 2026-06-09
> **Response to reviewer ngdw**
>
> We thank the reviewer for their helpful comments.
>
> We apologize for the delay in responding, which is due to one of the authors being OOO 3-8 June.
>
>
> > 1. Clarify the incremental contribution relative to Benavoli et   al. (2017), especially what is new beyond synthesis/tutorial  exposition.
>
> Note that Benavoli et al (2017) actually applied their paired comparisons only on the *aggregate* accuracies on each test fold in k-fold CV, not on individual test examples. So while the examples in
> sec 6. of the submission and in Benavoli et al. (2017) are similar they are not exactly the same; there is no CV in our case. We will clarify this in the revised m/s.
>
> Importantly, our argument (as on p. 1) is that the benchmark setting (development set, fixed test set) is the one that is used almost always in experimental comparisons, and is therefore the one that is
> the most important to address.
>
> We are not aware of any work that has applied paired test-example testing with a ROPE to the ML model comparison problem. This framing is developed by a careful review of the questions addressed (sec. 1) and issues of experimental design (sec. 3). This should also be regarded as a contribution to the paper, and is a response to the limitations of the standard point-null NHST paired tests (e.g. paired t-test, McNemar).
>
> These points will be clarified in the revised m/s.
>
> > 2. Add more discussion or sensitivity analysis for the ROPE choice, prior choices, and the proposed Bayesian McNemar/hierarchical  variants.
>
> Discussion of these issues will be expanded in the revised m/s.
>
> > 3. Either substantiate the claim that current ML practice is poor, or soften it as anecdotal motivation.
>
> We agree that the evidence presented that current ML practice is poor is anecdotal, and will soften the claim. In fact our observations are not just from one conference, but extend over many years. We have been pointing authors to Dietterich (1998) for many years, and indeed the motivation to work on this topic arises from frustrations about the SOTA in model comparison in the ML literature.
>
> > The likely audience is smaller than for a major new modeling method, and the methodological novelty is limited. However, statistically sound model comparison is central enough to empirical ML that some TMLR readers would benefit from the paper's findings and recommendations.
>
> Our ambition is to improve the SOTA in model comparison in ML. Empirical comparison of methods is very common, e.g. when proposing a new method. However, we agree that achieving this ambition will, of course, require much more than one paper, as discussed in sec. 8.

---

### Review · Reviewer_JrBe · 2026-06-03

**Summary Of Contributions:**

This paper discusses approaches to compare predictors and learning algorithms, advocating for the use of sound testing principles, and proposing a protocol to do this in practice. The paper argues for the importance of this endeavor, as the problem of comparison and testing arises in a large proportion of the literature in machine learning.

I believe that this is an important topic and I found the paper to be generally well written and easy to follow. I have a few comments that I hope will help improving the positioning and scope of the paper.

**Additional Comments:**

- Summary: Overall, this could be an interesting paper if there was a stronger motivation substantiated by a thorough analysis of the literature and some greater immediate practical implications of this work. From the methodological perspective, there is very little novelty, and this might be the biggest limitation overall.

**Audience:**

Yes

**Audience Explanation:**

I believe that the topic of sound statistical validation of predictors and learning algorithms is an important one for the ML and Stats communities.

**Claims And Evidence:**

No

**Claims Explanation:**

- Introduction: I believe it is important to make all the statements in the introduction (and in the paper at large) factual. At the moment, I do not agree with several of the statements made in the introduction because they do not reflect what I do in practice and I may have not come across papers with the specific flaws described there. Some examples:

"For example, of five papers we reviewed for a recent top-tier conference, only one had a satisfactory treatment of model comparison."

This is difficult to verify for any reader and it is a small sample. Would it not make sense to pick a large sample of papers (selected according to some sensible criteria (e.g., arXiv in the last year in a certain category) and do this analysis? I think this would significantly strengthen the introduction and the motivation.

"It is ironic that the ML field develops very sophisticated statistical models for its tasks, but often fails to use good statistical practice when doing model comparison."

Again, this is a bit of a strong statement that would need to be substantiated with some hard numbers - some readers that do their diligent work in reporting results with some form of statistical analysis may not agree. Also, the community has introduced checklists as a means to encourage sound statistical practices when reporting results, so it would be interesting to elaborate on this.

**Requested Changes:**

- Scope: I am a bit puzzled on the reasons behind the focus on

Q1: Comparison of predictors on a single task;
Q3: Comparison of learning algorithms across multiple tasks.

leaving aside Q2: Comparison of learning algorithms on a single task - maybe this is just an impression, but wouldn't it make more sense to say that addressing Q1 is useful for Q2, which in turn is useful for addressing Q3? And in Section 5 it seems that Q2 is discussed after all. Maybe some simple rewording can clear up this inconsistency.


- Novelty and related works: it seems that the majority of the methodological developments and ideas for comparisons appear in related works mentioned in the paper. I believe that a stronger text on the distinctive character of this paper is necessary. At the moment, it seems like there is little methodological novelty overall.


- Section 2: if the purpose of this section is to elaborate on appropriate loss functions for a given task, this is a bit too short and uninformative. If the purpose is to mention that there exist appropriate losses for any applications, then maybe this can be considered as something that the reader would know already, and the section could be turned into a short paragraph at the beginning of Sec 3.


- Section 3: the main methodological development taken from the literature rests on a Gaussian assumption on the loss (ANOVA model in equation 1). For certain losses, this assumption might be grossly violated, and it would be interesting to report on these cases and see how/when the proposed statistical analyses break down. It looks like rank tests are also proposed, and it would be interesting if the paper could elaborate more on these to make the paper more self-contained.


- I found the working examples quite nice at guiding the reader through the proposed protocol and that can illustrate the problems associated with a variety of statistical arguments. I wonder if it wouldn't be better to focus the paper on these working examples (possibly adding more), shortening the rest. But would this type of paper be a good fit for TMLR?


- I think it would be useful to discuss the sensitivity of the analyses with respect to the ROPE constant 0.1 in the interval [−0.1s,0.1s].


- I wonder if it wouldn't be appropriate to offer an implementation in the form of a software package to facilitate a wider adoption of the proposed testing methodology.


- One aspect I'm curious about is the possibility of obtain sound statistical comparisons in the case of models that are expensive to train and where it is absolutely impossible to run the models for different seeds/hyper-parameters. This is becoming an increasingly recurrent problem. Based on the ideas put forward in the paper, is there any practical advice that can be offered to the reader in these situations? I think this would widen the interest in the ideas presented in this paper.

---

> ### Author Response · Authors · 2026-06-09
> **Response to reviewer JrBe -- part 1**
>
> We thank the reviewer for their helpful comments.
>
> We apologize for the delay in responding, which is due to one of the
> authors being OOO 3-8 June.
>
>
> > Introduction: I believe it is important to make all the statements in
> the introduction (and in the paper at large) factual. At the moment, I
> do not agree with several of the statements made in the introduction
> because they do not reflect what I do in practice and I may have not
> come across papers with the specific flaws described there. Some
> examples: "For example, of five papers we reviewed for a recent
> top-tier conference, only one had a satisfactory treatment of model
> comparison."
>
> and
>
> > "It is ironic that the ML field develops very sophisticated
> statistical models for its tasks, but often fails to use good
> statistical practice when doing model comparison."
> Again, this is a bit of a strong statement that would need to be
> substantiated with some hard numbers - some readers that do their
> diligent work in reporting results with some form of statistical
> analysis may not agree. Also, the community has introduced checklists
> as a means to encourage sound statistical practices when reporting
> results, so it would be interesting to elaborate on this.
>
> Our view about that the ML field often fails to use good statistical practice when doing model comparison comes not from one conference reviewing session (as it may appear in the text), but from experience of poor practice over many years. Indeed the motivation to work on
> this topic arises from frustrations about the SOTA in model comparison in the ML literature.  However, we agree that this is anecdotal evidence, and so will tone down the statements the referee refers to above in the revised m/s.
>
> **Requested Changes:**
>
> >Scope: I am a bit puzzled on the reasons behind the focus on
> Q1: Comparison of predictors on a single task; Q3: Comparison of
> learning algorithms across multiple tasks. leaving aside Q2: Comparison of learning algorithms on a single task - maybe this is just an impression, but wouldn't it make more sense to
> say that addressing Q1 is useful for Q2, which in turn is useful for
> addressing Q3? And in Section 5 it seems that Q2 is discussed after
> all. Maybe some simple rewording can clear up this inconsistency.
>
> Q2 is discussed in para 4 of sec. 3, but in the context of multiple *independent* training sets and a common test set. But this setting, which is attractive for analysis, is hardly ever used in practice.
> Instead, resampling methods such as cross validation are used, but this complicates the analysis. Benavoli et al (2017, p 10) note that the correlation parameter $\rho$ needed in their Bayesian
> correlated t-test is not identifiable, and use the heuristic from Nadeau and Bengio (2003).
>
> On a similar issue, Demsar (2006, p. 5) comments: "Furthermore, the problem of correct statistical tests for comparing classifiers on a single data set is not related to the comparison on multiple data sets in the sense that we would first have to solve the former problem in order to tackle the latter. Since running the algorithms on multiple data sets naturally gives a sample of independent measurements, such comparisons are even simpler than comparisons on a single data set."
>
> So although the view espoused above about Q2 being useful for addressing Q3 seems reasonable, it turns out that it is not really correct. We will add explanatory text on this point in the revised
> m/s.
>
> > Novelty and related works: it seems that the majority of the
> methodological developments and ideas for comparisons appear in
> related works mentioned in the paper. I believe that a stronger text
> on the distinctive character of this paper is necessary. At the
> moment, it seems like there is little methodological novelty overall.
>
> Note that Benavoli et al (2017) actually applied their paired comparisons only on the *aggregate* accuracies on each test fold in k-fold CV, not on individual test examples. So while the examples in
> sec 6. of the submission and in Benavoli et al. (2017) are similar they are not exactly the same; there is no CV in our case. We will clarify this in the revised m/s.
>
> Importantly, our argument (as on p. 1) is that the benchmark setting (development set, fixed test set) is the one that is used almost always in experimental comparisons, and is therefore the one that is
> the most important to address.
>
> We are not aware of any work that has applied *paired test-example testing with a ROPE* to the ML model comparison problem. This framing is developed by a careful review of the questions addressed (sec. 1) and issues of experimental design (sec. 3). This should also be
> regarded as a contribution to the paper, and is a response to the limitations of the standard point-null NHST paired tests (e.g. paired t-test, McNemar).
>
> These points will be clarified in the revised m/s.
>
> [response continues below, due to 5000 chars limit]

---

> > ### Author Response · Authors · 2026-06-09
> > **Response to reviewer JrBe -- part 2**
> >
> > > Section 2: if the purpose of this section is to elaborate on
> > appropriate loss functions for a given task, this is a bit too short
> > and uninformative. If the purpose is to mention that there exist
> > appropriate losses for any applications, then maybe this can be
> > considered as something that the reader would know already, and the
> > section could be turned into a short paragraph at the beginning of Sec
> > 3.
> >
> > We agree sec. 2 is short, but it is logically distinct from sec. 3 which is about fixed and random factors and experimental design. We thus prefer to keep the division as is.
> >
> > > Section 3: the main methodological development taken from the
> > literature rests on a Gaussian assumption on the loss (ANOVA model in
> > equation 1). For certain losses, this assumption might be grossly
> > violated, and it would be interesting to report on these cases and see
> > how/when the proposed statistical analyses break down. It looks like
> > rank tests are also proposed, and it would be interesting if the paper
> > could elaborate more on these to make the paper more self-contained.
> >
> > The Gaussian assumption on the loss can be relaxed.  The discussion in sec 5.3 (last para of p 8) explains that the distribution of the $d_j$'s can be explored, e.g. with a Q-Q plot, and that a robustified analysis can be carried out, e.g. with a Student-t distribution.  In
> > fact such robustified analyses are carried out in sec 6.1 for the LgRSVM and LgRMLP examples.
> >
> > > I found the working examples quite nice at guiding the reader through
> > the proposed protocol and that can illustrate the problems associated
> > with a variety of statistical arguments. I wonder if it wouldn't be
> > better to focus the paper on these working examples (possibly adding
> > more), shortening the rest. But would this type of paper be a good fit
> > for TMLR?
> >
> > We certainly agree that the worked examples are an important part of the paper, and are intended to illustrate good practice. However, one needs to explain the logic of the choices made, and that is what the preceding sections do.
> >
> > > I think it would be useful to discuss the sensitivity of the analyses
> > with respect to the ROPE constant 0.1 in the interval [−0.1s,0.1s].
> >
> > We agree and this will be expanded in the revised m/s.
> >
> > > I wonder if it wouldn't be appropriate to offer an implementation in
> > the form of a software package to facilitate a wider adoption of the
> > proposed testing methodology.
> >
> > Absolutely, and in sec. 5.7 we mention our repository. However for double-blind review this link has been removed.